# Neuron type-specific increase in lamin B1 contributes to nuclear dysfunction in Huntington's disease

Rafael Alcalá-Vida[1,2,3,†,‡], Marta Garcia-Forn[1,2,3,†,§], Carla Castany-Pladevall[1,2,3], Jordi Creus-Muncunill[1,2,3], Yoko Ito[4], Enrique Blanco[5], Arantxa Golbano[1,2,3], Kilian Crespí-Vázquez[1,2,3], Aled Parry[6], Guy Slater[4], Shamith Samarajiwa[7], Sandra Peiró[8], Luciano Di Croce[5,9,10], Masashi Narita[4] & Esther Pérez-Navarro[1,2,3,*]

## Abstract

Lamins are crucial proteins for nuclear functionality. Here, we provide new evidence showing that increased lamin B1 levels contribute to the pathophysiology of Huntington's disease (HD), a CAG repeat-associated neurodegenerative disorder. Through fluorescence-activated nuclear suspension imaging, we show that nucleus from striatal medium-sized spiny and CA1 hippocampal neurons display increased lamin B1 levels, in correlation with altered nuclear morphology and nucleocytoplasmic transport disruption. Moreover, ChIP-sequencing analysis shows an alteration of lamin-associated chromatin domains in hippocampal nuclei, accompanied by changes in chromatin accessibility and transcriptional dysregulation. Supporting lamin B1 alterations as a causal role in mutant huntingtin-mediated neurodegeneration, pharmacological normalization of lamin B1 levels in the hippocampus of the R6/1 mouse model of HD by betulinic acid administration restored nuclear homeostasis and prevented motor and cognitive dysfunction. Collectively, our work points increased lamin B1 levels as a new pathogenic mechanism in HD and provides a novel target for its intervention.

**Keywords** chromatin accessibility; LAD; nuclear morphology; nuclear permeability; R6/1 mouse

**Subject Categories** Molecular Biology of Disease; Neuroscience

## Introduction

Lamins are type V intermediate filaments that together with lamin-binding proteins are embedded into the inner nuclear membrane and constitute the nuclear lamina (de Leeuw *et al*, 2018). This family of proteins is classified into two subgroups: A-type lamins (lamins A and C), encoded by the *LMNA* gene, and B-type lamins (lamins B1 and B2), encoded by *LMNB1* and *LMNB2* genes, respectively (de Leeuw *et al*, 2018). It was long thought that their only function was to provide a structural support to the nuclear envelope membrane, but evidence indicates that they are involved in a wide variety of cell functions and processes, including DNA replication, transcription, chromatin organization, and nucleus–cytoplasm interaction (Hozak *et al*, 1995). While lamins A and C are expressed exclusively in differentiated cells, lamin B is present in almost all cell types independently of their differentiation state (Verstraeten *et al*, 2007). This suggests that B-type lamins are essential for the survival of mammalian cells (Harborth *et al*, 2001).

Alterations in lamins content or structure lead to a particular type of nuclear envelopathies called laminopathies (Schreiber & Kennedy, 2013). While many laminopathies are associated with mutations in *LMNA* gene (Schreiber & Kennedy, 2013), only two have been associated with alterations in lamin B: the autosomal

1 Departament de Biomedicina, Facultat de Medicina i Ciències de la Salut, Institut de Neurociències, Universitat de Barcelona, Barcelona, Catalonia
2 Institut d'Investigacions Biomèdiques August Pi i Sunyer (IDIBAPS), Barcelona, Catalonia
3 Centro de Investigación Biomédica en Red sobre Enfermedades Neurodegenerativas (CIBERNED), Madrid, Spain
4 Cancer Research UK Cambridge Institute, University of Cambridge, Cambridge, UK
5 Centre for Genomic Regulation (CRG), The Barcelona Institute of Science and Technology, Barcelona, Spain
6 Epigenetics Programme, The Babraham Institute, Cambridge, UK
7 MRC Cancer Unit, Hutchison/MRC Research Centre, University of Cambridge, Cambridge, UK
8 Vall d'Hebron Institute of Oncology, Barcelona, Spain
9 Universitat Pompeu Fabra (UPF), Barcelona, Spain
10 ICREA, Barcelona, Spain
*Corresponding author. Tel: +34 93 4035284; E-mail: estherperez@ub.edu
†These authors contributed equally to this work
‡Present address: Laboratory of Cognitive and Adaptive Neuroscience, UMR 7364 (CNRS/Strasbourg University), Strasbourg, France
§Present address: Seaver Autism Center for Research and Treatment, Icahn School of Medicine at Mount Sinai, New York, NY, USA

dominant leukodystrophy (Padiath, 2019) and the acquired partial lipodystrophy (Hegele et al, 2006) caused by LMNB1 and LMNB2 mutations, respectively. Interestingly, in the last few years, lamin B alterations have also been found in neurodegenerative disorders such as Parkinson's disease (PD) and Alzheimer's disease (AD) (Liu et al, 2012; Frost, 2016). Strikingly, two cellular functions in which lamin B plays a critical role, RNA nuclear exportation (Gasset-Rosa et al, 2017) and nuclear pore complex organization (Grima et al, 2017), are altered in Huntington's disease (HD), an autosomal dominant neurodegenerative disorder caused by an inherited CAG repeat expansion in the exon 1 of the huntingtin (htt) gene (HDCRG, 1993). This mutation results in the lengthening of the polyglutamine chain at the amino terminus of the huntingtin (Htt) protein inducing self-association and aggregation. Consequently, mutant Htt (mHtt) loses its biological functions and becomes toxic (Ross & Poirier, 2004). In HD, medium-sized spiny neurons (MSNs), the GABAergic output projection neurons that account for the vast majority (90–95%) of all striatal neurons, are mainly affected. Although motor symptoms are the most prominent, psychiatric alterations and cognitive decline appear first in HD patients and become more evident as the disease progresses. Cognitive deficits are related to the dysfunction of the corticostriatal pathway and the hippocampus and, together with motor deficits, have been replicated in most HD mouse models (Puigdellívol et al, 2016).

Molecular mechanisms leading to neuronal dysfunction in HD remain to be elucidated. Previous results from our laboratory suggested that decreased levels of the pro-apoptotic kinase PKCδ would lead to an aberrant accumulation of lamin B (Rué et al, 2014) which, in turn, could have a significant influence in the nuclear lamina structure and function (Lin & Fu, 2009; Ferrera et al, 2014). Therefore, here we sought to deeply characterize the impact of lamin alterations in HD brain at physiological (studying nuclear lamina morphology and nucleocytoplasmic transport), transcriptomic (by generating RNA-sequencing (RNA-seq) data), and epigenetic (analyzing lamin chromatin binding and chromatin accessibility) levels by using the R6/1 transgenic mouse model of HD and human post-mortem brain samples.

## Results

### Lamin B levels are increased in a region-specific manner in HD brain

Lamin B1, lamin B2, and lamin A/C protein levels were analyzed in the striatum, cortex, and hippocampus of wild-type and R6/1 mice, a transgenic mouse model of HD overexpressing the exon 1 of the human mHtt (Mangiarini et al, 1996), at different ages. Western blot analysis revealed an increase in lamin B1 (Fig 1A) and lamin B2 (Fig 1B) levels in all three regions from early disease stages in R6/1 mice, whereas lamin A/C protein levels remained unchanged until 30 weeks of age in the striatum and hippocampus (Fig EV1). Since the most important alterations were found in lamin B isoforms, we investigated whether such an increase was reproducible in the brain of HD patients. Western blot analysis revealed that lamin B1 levels were significantly higher, in comparison with levels in non-affected individuals, in the putamen of HD patients at Vonsattel (VS) grade III–grade IV, and in the frontal cortex of HD patients at grade

I–grade II and grade III–grade IV (Fig EV2A). Unexpectedly, no significant changes were found within the hippocampus of HD patients at any disease stage. On the other hand, lamin B2 protein levels were only increased in the frontal cortex of HD patients (Fig EV2B). Consequently, only lamin B1 levels are consistently affected in the brain of R6/1 mice and HD patients.

Previous studies have shown lamin alterations in the nucleus of aged human fibroblasts (Freund et al, 2012) and keratinocytes (Dreesen et al, 2013). In order to discard that our observations were due merely to the aging process itself, we analyzed the correlation between lamin B1 levels and age in human samples from the putamen, cortex, and hippocampus. We observed no correlation between age and lamin B1 levels in any of the brain regions analyzed (Appendix Fig S1A–C). In addition, the distribution of ages showed that control and HD samples were age-matched (Appendix Fig S1D). Thus, our results indicate that alterations in lamin B1 levels are occurring because of the HD pathology itself. In contrast to other studies showing that some altered mechanisms, such as transcriptomic dysregulation, are dependent on the CAG repeat length (Langfelder et al, 2016), we did not observe a correlation between the number of CAG repeats and lamin B1 protein levels in the putamen, hippocampus, and cortex of HD patients (Appendix Fig S2). Thus, lamin B1 alteration is dependent on the pathological stage rather than on the original number of inherited CAGs.

In an attempt to investigate the molecular mechanisms leading to the increase in lamin B1 levels, and considering our previous results showing a possible link between decreased PKCδ and increased lamin B levels in HD brain (Rué et al, 2014), we knocked down PKCδ in striatal cells expressing mHtt. Once the efficiency of PKCδ siRNA was analyzed by Western blot (Fig EV3A), wild-type striatal cells (STHdh$^{Q7/Q7}$) were co-transfected with exon 1 encoded N-terminal Htt with 94 glutamines fused to cyan-fluorescent protein (CFP) (N-mHtt-CFP) and a PKCδ siRNA or a scramble siRNA as a control. Lamin B1 levels were analyzed by immunocytochemistry 24 h after transfection. We observed that lamin B1 levels were increased in the nucleus of striatal cells transfected with N-mHtt-CFP in the PKCδ siRNA condition as compared to those cells transfected with N-mHtt-CFP plus scramble siRNA (Fig EV3B and C). Moreover, we analyzed PKCδ levels in striatal and hippocampal samples from the same R6/1 mice used to analyze lamin B1 levels, and we observed a significant correlation between the reduction in PKCδ and the increase in lamin B1 levels in the striatum, but not in the hippocampus (Fig EV3D). Therefore, our results suggest that decreased PKCδ levels could be involved in the accumulation of lamin B1 in R6/1 mouse striatum, but not in the hippocampus.

### Lamin B1-increased levels are mainly localized in neurons

To address the cell type specificity of the lamin B1 increase, we performed NeuN and lamin B1 co-immunostaining (with or without GFAP), in brain sections obtained from 30-week-old R6/1 mice and from HD patients at different stages of the disease, and their corresponding controls. Observation of confocal z-stacks images from the striatum of R6/1 mice showed a strong increase in lamin B1 signal, which was more prominent in NeuN-positive nuclei (Fig 2A). These nuclei showed nuclear lamina invaginations and lamin B1 delocalization within the nucleoplasm, resulting in altered morphological

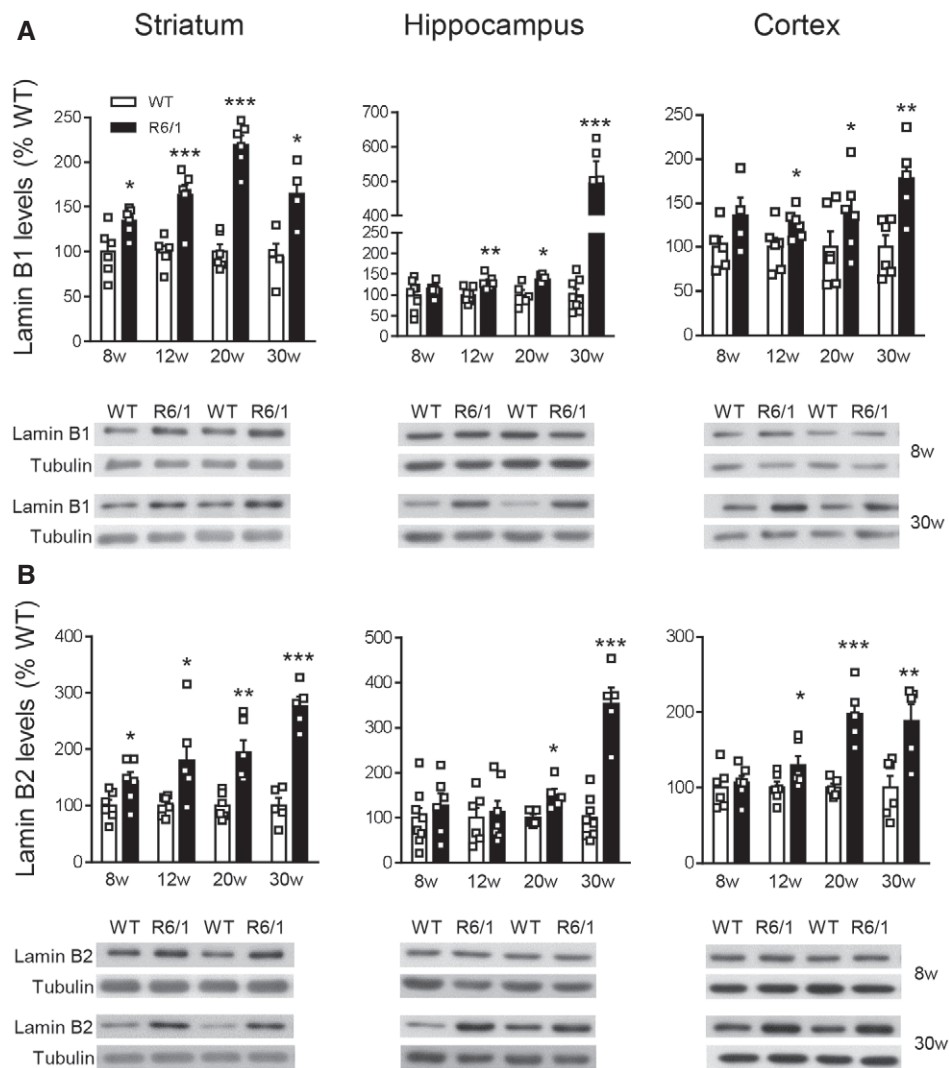

**Figure 1. Lamin B1 and lamin B2 are differentially increased in R6/1 mouse brain.**

Lamin B1 and lamin B2 were analyzed by Western blot in R6/1 mice at different stages of the disease progression (w: weeks) and their corresponding controls (WT: wild-type littermates).

A Quantification and representative immunoblots of lamin B1 in the striatum (8w, 12w, and 20w: $N = 6$ for both genotypes; 30w: $N = 4$ and 5 for WT and R6/1 mice, respectively), hippocampus (8w: $N = 8$ and 6 for WT and R6/1 mice, respectively; 12w: $N = 7$ and 6 for WT and R6/1 mice, respectively; 20w: $N = 5$ and 6 for WT and R6/1 mice, respectively; 30w: $N = 8$ and 5 for WT and R6/1 mice, respectively), and cortex (8w: $N = 5$ and 4 for WT and R6/1 mice, respectively; 12w and 20w: $N = 6$ for both genotypes; 30w: $N = 6$ and 5 for WT and R6/1 mice, respectively).

B Quantification and representative immunoblots of lamin B2 in the striatum (8w, 12w, and 20w: $N = 6$ for both genotypes, 30w: $N = 5$ for both genotypes), hippocampus (8w: $N = 8$ and 6 for WT and R6/1 mice, respectively; 12w: $N = 6$ and 7 for WT and R6/1 mice, respectively; 20w: $N = 5$ for both genotypes; 30w: $N = 8$ and 5 for WT and R6/1 mice, respectively), and cortex (8w and 12w: $N = 6$ for both genotypes; 20w: $N = 5$ for both genotypes; 30w: $N = 8$ and 5 for WT and R6/1 mice, respectively).

Data information: Data are expressed as a percentage of controls. Each point corresponds to the value from an individual sample. Bars represent the mean $\pm$ SEM. *$P < 0.05$, **$P < 0.01$, and ***$P < 0.001$ as compared with WT mice (two-tailed unpaired Student's $t$-test). Tubulin was used as loading control. Exact $P$ values are reported in Appendix Table S3.

Source data are available online for this figure.

parameters as compared to wild-type mice nuclei (Fig EV4). Furthermore, and similarly to what we observed for GFAP-positive cells, nucleus from striatal oligodendrocytes did not show alterations in lamin B1 levels nor in morphology (Appendix Fig S3). In line with that, neurons from the putamen of HD patients displayed lamin B1 signal delocalization within the nucleus, while lamin B1

was not altered in the nuclei of GFAP-positive cells (Fig 2B and C). In the hippocampus of R6/1 mice, the dentate gyrus (DG) and the Cornu ammonis 1 (CA1) regions presented the highest increase in lamin B1 signal (Fig 3A). Moreover, CA1 neuronal nuclei displayed morphological alterations and lamin B1 protein delocalization from the nuclear periphery, as observed in 3D reconstructions of z-stack

lamin B1 images (Fig 3B). Interestingly, at early disease stages (12 weeks), these changes appeared to be restricted to the CA1 region (Appendix Fig S4). Strikingly, these morphological alterations seemed to be independent to the presence of mHtt inclusions as indicated by the EM48 immunostaining in the R6/1 mice striatal and CA1 hippocampal nuclei (Appendix Fig S5). Overall, these results indicated a cell-type-specific increase in lamin B1 levels in response to mHtt in co-occurrence with lamin B1 delocalization and nuclear lamina morphological alterations that seemed to be independent of the presence of mHtt inclusions.

## Cell type-specific nuclear morphology alterations correlate with increased lamin B1 levels

To further confirm our observations indicating increased lamin B1 protein levels and nuclear morphology alteration in specific cell populations, fluorescence-activated nuclear suspension imaging (FANSI) method was developed by combining nuclear isolation from brain tissue with immunostaining (Benito *et al*, 2015) and the recently developed imaging flow cytometry (Barteneva & Vorobjev, 2016), allowing the acquisition of individual nucleus images. By combining antibodies against Ctip2 and NeuN, we discerned between MSNs (Ctip2$^+$/NeuN$^+$), which represent around 90% of total striatal neurons (Kemp & Powell, 1971), striatal interneurons (Ctip2$^-$/NeuN$^+$), and glial cells (Ctip2$^-$/NeuN$^-$; Appendix Fig S6A). As shown in Fig 4, we detected an increase in lamin B1 levels and altered nuclear morphology in 30-week-old R6/1 MSNs (Fig 4A), but not in striatal interneurons (Fig 4B) nor in glial cells (Appendix Fig S7), supporting our previous observations (see Fig 2 and Appendix Fig S3). Nuclear area and total number of counted nuclei were not altered in comparison with wild-type mice, in agreement with the lack of neuronal death observed in R6/1 mouse brain (Francelle *et al*, 2014). In addition, we confirmed that these alterations were independent to the presence of mHtt inclusions. As shown in Fig 4C, alterations in lamin B1 levels and nuclear morphology similarly occur in striatal neuronal nuclei with or without mHtt inclusions. In the putamen of HD patients, the number of Ctip2$^+$/NeuN$^+$ nuclei examined by FANSI was extremely low (10 neuronal nuclei in average for each sample) what made it difficult to reach a conclusion (Appendix Fig S8A). Therefore, we decided to analyze lamin B1 intensity and circularity by immunohistochemistry. We observed that lamin B1 intensity was increased only in the nuclei of MSNs from VS III-IV patients, in correlation with altered nuclear morphology (Fig 4D). In addition, and accordingly to results obtained in R6/1 mice striatum, FANSI analysis showed no alterations in the nuclei of striatal glial cells from HD patients in comparison with control

individuals (Appendix Fig S8B). In the hippocampus, to distinguish between DG and CA1 neuronal nuclei, we used antibodies against Ctip2 and Prox1 (Appendix Fig S6B). In 30-week-old R6/1 mouse hippocampus, we detected increased lamin B1 levels only in CA1 neuronal nuclei (Ctip2$^+$/ Prox1$^-$) in concomitance with morphological alterations, while neuronal nuclei from DG (Ctip2$^+$/ Prox1$^+$) were relatively spared. Furthermore, no alterations were found in the total number of nuclei or in nuclear area (Fig 4E). Altogether, these results suggest that increased lamin B1 levels contribute to nuclear morphology alterations in specific neuronal populations in HD brain. To test this, striatal primary cultures were transfected with a plasmid expressing lamin B1-mApple or mApple as a control, and nuclear morphology was examined 24 h after transfection by confocal microscopy. We observed that lamin B1 overexpression in cultured striatal neurons induced a dramatic alteration of nuclear morphology (Fig 5A and B) and chromatin condensation (Fig 5B).

## Increased lamin B1 levels correlate with alterations in nuclear permeability in MSNs and CA1 hippocampal neurons from R6/1 mice

Our previous results showed that striatal MSNs and CA1 hippocampal neuronal nuclei are preferentially affected by altered lamin B1 levels. Since alterations in nuclear architecture lead to changes in nuclear permeability (Hatch & Hetzer, 2014), we wondered whether increased lamin B1 levels could contribute to nuclear transport abnormalities. To address this hypothesis, we performed fluorescence recovery after photobleaching (FRAP) experiments in isolated 30-week-old wild-type and R6/1 mice neuronal nuclei by using 20 kDa FITC-dextran (Fig 6A). We detected that half-time of recovery (t1/2) in R6/1 striatal MSNs (Fig 6B) and CA1 hippocampal (Fig 6C), but not DG (Fig 6D), nuclei was slower than in wild-type mice. Bleaching percentage and maximum fluorescence recovered (plateau) did not differ between genotypes (Appendix Fig S9A and B). As a validation, FRAP was performed in the background, where any of the analyzed parameters was found altered (Appendix Fig S9C). Altogether, our results show an alteration in the passive diffusion of dextran into neuronal nuclei containing greater lamin B1 protein levels, suggesting that nucleocytoplasmic passive transport abnormalities are linked to lamin B1 alterations in a cell type-dependent manner.

## Lamin B1 chromatin binding is impaired in R6/1 mice hippocampus

Lamin B1 is classically associated with large heterochromatin domains called lamin-associated domains (LADs), characterized by

---

**Figure 2. Lamin B1 distribution in the striatum of R6/1 mouse and in the putamen of HD patients.**

A   Mouse brain tissue was processed for immunohistochemistry by combining anti-lamin B1 (red) and anti-NeuN (green) antibodies. Representative images (maximal Z-projections) showing the distribution of lamin B1 in the striatum of 30-week-old wild-type (WT) and R6/1 mice. Yellow arrowheads show co-localization between lamin B1 and NeuN, and white arrowheads show NeuN negative lamin B1-positive nuclei. Scale bar 50 and 25 μm for low and high magnification, respectively.

B   Lamin B1 distribution in human putamen was analyzed by immunohistochemistry. Antibody against lamin B1 (red) was combined with DAPI Fluoromount-G (blue) to label nuclei. Representative images show the distribution of lamin B1 in the putamen of non-affected individuals (CTL) and HD patients at different stages of the disease (VS II-IV: Vonsattel grades). Yellow and white arrowheads indicate MSNs and glial cells, respectively. Scale bar 50 and 20 μm for low and high magnification, respectively.

C   The distribution of lamin B1 in the putamen of HD patients was analyzed by immunohistochemistry. Anti-lamin B1 antibody (red) was combined with anti-GFAP antibody (green), and nuclei were labeled with DAPI Fluoromount-G (blue). Representative images show the distribution of lamin B1 at Vonsattel grade III. Yellow arrowheads indicate MSNs, and white arrowheads indicate GFAP-positive cells. Scale bar 10 μm.

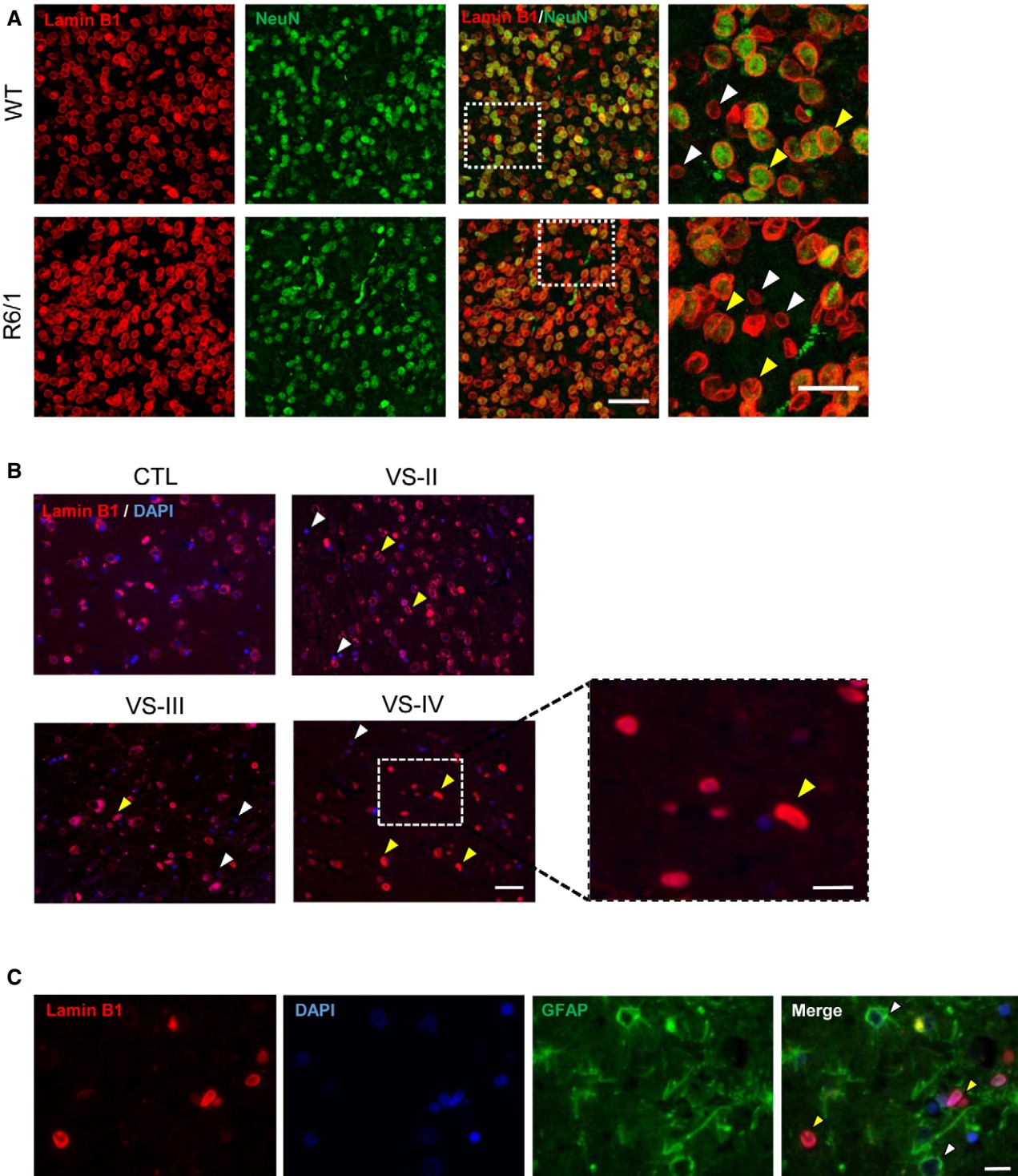

**Figure 2.**

low gene expression levels (Belmont *et al*, 1993). Recently, however, LADs have been linked to actively transcribed euchromatic regions (Pascual-Reguant *et al*, 2018). Therefore, we set out to study whether increased lamin B1 levels in HD brain could alter their chromatin-binding landscape. For that, we generated lamin B1 chromatin immunoprecipitation and sequencing (ChIP-seq) data in

30-week-old wild-type and R6/1 mice hippocampus (age and region showing the highest increase in lamin B1 levels). We immunoprecipitated lamin B1 and verified that both heterochromatin and euchromatin fractions were efficiently sonicated (Appendix Fig S10A and B). This indicated that all lamin B1-bound regions should be captured in our ChIP-seq experiments. We ran the EED peak

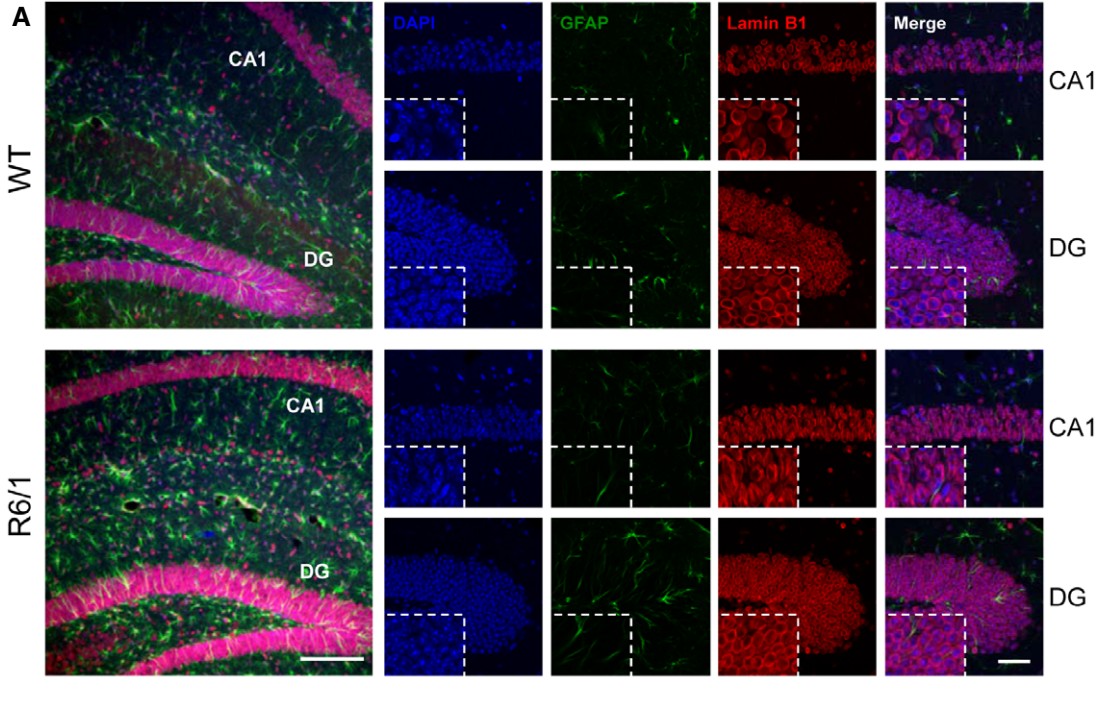

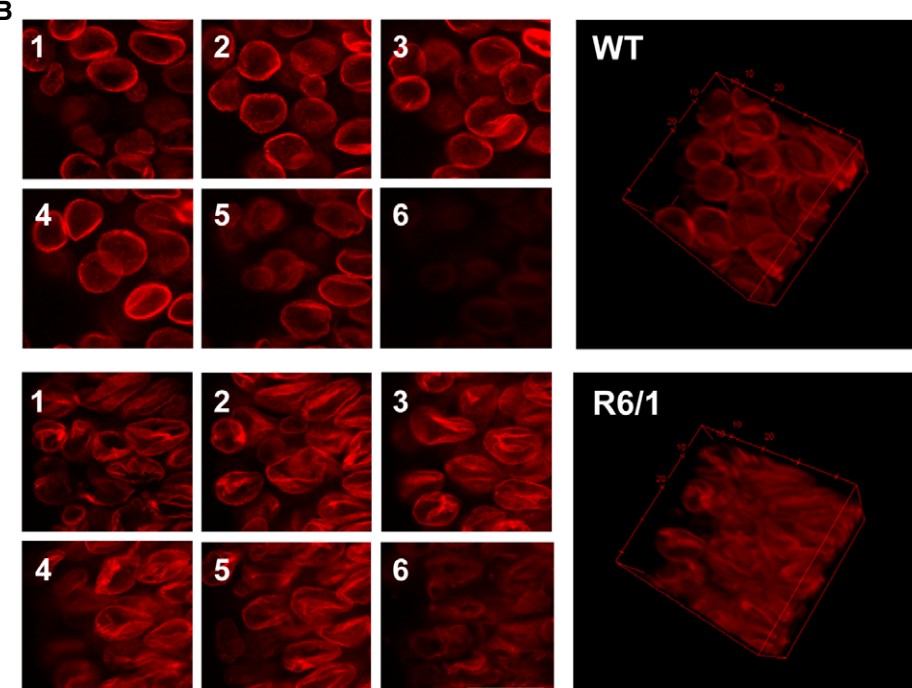

**Figure 3. Lamin B1 distribution in the hippocampus of R6/1 mouse.**

Hippocampal sections from 30-week-old wild-type (WT) and R6/1 mice were labeled with anti-lamin B1 antibody (red), and anti-GFAP antibody (green) and nuclei were labeled with DAPI Fluoromount-G (blue).

A Representative images showing the distribution of lamin B1. On the left, images showing maximal Z-projection. Scale bar 150 µm. Small images correspond to CA1 and DG magnified images showing independent DAPI, GFAP, and lamin B1 channels, and merge, of representative confocal Z-stack images. Scale bar 50 µm.

B Left, representative images showing the distribution of lamin B1 in the nuclei of CA-1 hippocampal neurons from WT and R6/1 mice along different Z-axis planes (1-6). On the right, a 3D reconstruction of the Z-stack confocal images generated using ImageJ. Scale bar 20 µm.

calling tool (Lund *et al*, 2014) and identified 145 and 166 LADs in wild-type and R6/1 mice hippocampus, respectively (Fig 7A and B), showing a genotype-dependent component in our lamin B1 ChIP-seq data (Appendix Fig S10C). These numbers were consistently found in triplicates of each experiment (Appendix Fig S10D and E), and the regions overlapped with LADs previously identified in neural progenitor cells using DamID (Peric-Hupkes *et al*, 2010) (Appendix Fig S10F). Our LADs were depleted of H3K9ac and CTCF and highly enriched in H3K9me3, which are major features of canonical gene-silencing LADs (Appendix Fig S10G). We observed a lower average size and lamin B1 binding in R6/1 mice-specific LADs (Fig 7A and B). Interestingly, while most of the regions identified by EDD were common between wild-type and R6/1 mice, a small subset of them were specifically identified in one of the genotypes (Fig 7C). We found regions specific from R6/1 mouse, but they were equally enriched in lamin B1 in both genotypes, suggesting an arte-factual origin from the EDD tool. For wild-type mice-specific regions, however, we observed a clear reduction in lamin B1 bind-ing in R6/1 mice (Fig 7C). Genes located within common LADs between wild-type and R6/1 mice were enriched in terms mostly related to "olfactory sensory perception" and "keratinization" (Fig 7 D). However, genes within wild-type mice exclusively identified regions showed a strong differential functional signature, being mostly enriched in genes related to "nucleosome assembly" (Fig 7 D). In line with this, nuclear fractionation clearly showed a reduc-tion in the proportion of lamin B1 bound to chromatin, with a paral-lel accumulation of lamin B1 protein within the nucleoplasm (Fig 7E). Altogether, these results suggest that alterations in lamin B1 protein levels and localization in R6/1 mice hippocampus lead to changes in the genome-wide map of LADs, which ultimately could affect the expression and accessibility of certain genes.

### Chromatin accessibility, gene transcription, and LAD organization in R6/1 mice hippocampus

To study the impact of lamin B1 chromatin-binding alterations in chromatin state and gene expression, we analyzed chromatin acces-sibility and gene expression levels by generating assay for trans-posase-accessible chromatin and parallel sequencing (ATAC-seq; Fig 8A and Appendix Fig S11A) and RNA-seq data using 30-week-old R6/1 mice hippocampus. We identified a similar number of ATAC-seq peaks in wild-type and R6/1 mice hippocampus ($260,284 \pm 4,559$ and $254,672 \pm 9,030$, respectively) by using three independent biological replicates, indicating no massive changes in chromatin accessibility between genotypes. Differential peak acces-sibility analysis showed a high genotype-dependent component in our ATAC-seq data (Appendix Fig S11B), and identified 1,304 and 803 regions with gained or lost accessibility, respectively, in R6/1 mice hippocampus (adjusted *P*-value < 0.05). These regions were predominantly distal regulatory elements localized at intronic and intergenic regions (Appendix Fig S11D). Motif analysis identified EGR1/2 and NEUROD2 as centrally transcription factors enriched in each set of differential accessible peaks (Appendix Fig S11C). To gain insight into functional relevance of these changes, differential accessible regions were annotated to the closest transcription start site (TSS). Genes showing a loss of accessibility in R6/1 mice displayed a clear neuronal signature, with enriched terms such as "positive regulation of synapse" or "chemical synaptic

transmission", while genes associated with an increase in chromatin accessibility were mostly associated with developmental- and tran-scriptional-related terms (Fig 8B). The transcriptome of both geno-types ($n = 9$) clearly differed (as shown by PCA in Appendix Fig S11E). We found 2,145 up-regulated and 2,280 down-regulated genes (adjusted *P*-value < 0.001) in R6/1 with respect to wild-type mice hippocampus. Gene ontology analysis of differentially expressed genes showed high homology with the one found in our ATAC-seq data, with a predominance of neuronal-related and tran-scriptional-related terms for down- and up-regulated genes, respec-tively (Appendix Fig S11F). In addition, we found a substantial overlap with previously identified sets of altered expressed genes in other HD mouse models (Appendix Fig S11G) (Langfelder *et al*, 2016; Hervás-Corpión *et al*, 2018). As expected, we found that only genes showing the highest transcriptional dysregulation (adjusted *P*-value < 0.001, |fold change| > 2) displayed significant changes in chromatin accessibility at their TSS as compared with genes only fil-tered according to their adjusted *P*-value (Fig 8C and Appendix Fig S11H). However, when analyzing transcriptional changes associated with identified differential accessible regions, a clear correlation was observed (Fig 8D), suggesting that distal regulatory element accessi-bility better accounts for transcriptional dysregulation in R6/1 mice hippocampus.

When focusing on LADs reported in both genotypes, as expected, we found a particular enrichment in genes with low transcriptional rate (Fig 9A). However, when we analyzed LADs specifically found in wild-type mice or common between both genotypes (Fig 7C), minor changes were found either at transcriptional level (Fig 9B) or in terms of chromatin accessibility (Fig 9C), indicating that loss of lamin B1 binding in R6/1 mice hippocampal cells does not lead to a global transcriptional induction in these regions. Additionally, we analyzed the presence of genes differentially expressed within the set of genes specifically found in wild-type mice LADs (lost in R6/1 mouse; see Fig 7C) or in wild-type and R6/1 mice common LADs (Fig 9D). We found that genes differentially expressed between wild-type and R6/1 mice were more enriched in wild-type-specific LADs than in wild-type and R6/1 common LADs (up-regulated: 119/ 1242 in wild-type-specific LADs versus 142/3654 in common LADs; down-regulated: 110/1242 in wild-type-specific LADs versus 198/ 3654 in common LADs), indicating that loss of lamin B1 chromatin binding in R6/1 mice hippocampal cells could lead to chromatin reorganization affecting genes within these regions. However, when focusing on highly dysregulated genes (|fold change| > 2), this enrichment was only observed for down-regulated genes (down-regulated: 10/1,242 in wild-type-specific LADs versus 21/3,654 in common LADs; up-regulated: 3/1,242 in wild-type-specific LADs versus 9/3,654 in common LADs). Furthermore, we analyzed whether regions with differential accessibility between wild-type and R6/1 mice displayed changes in lamin B1 chromatin binding. Not surprisingly, ATAC-seq regions were generally depleted of lamin B1 occupancy (Fig 9E, left panel). However, when comparing regions showing increased or decreased chromatin accessibility in R6/1 mice to unchanged regions, a highest occupancy of lamin B1 was observed in the first, especially in regions with decreased acces-sibility (Fig 9E, right panel). Interestingly, a significant decrease in lamin B1 occupancy was observed in regions with increased accessi-bility in R6/1 mice, suggesting that lamin B1 chromatin-binding impairment could indeed lead to localized increase in chromatin

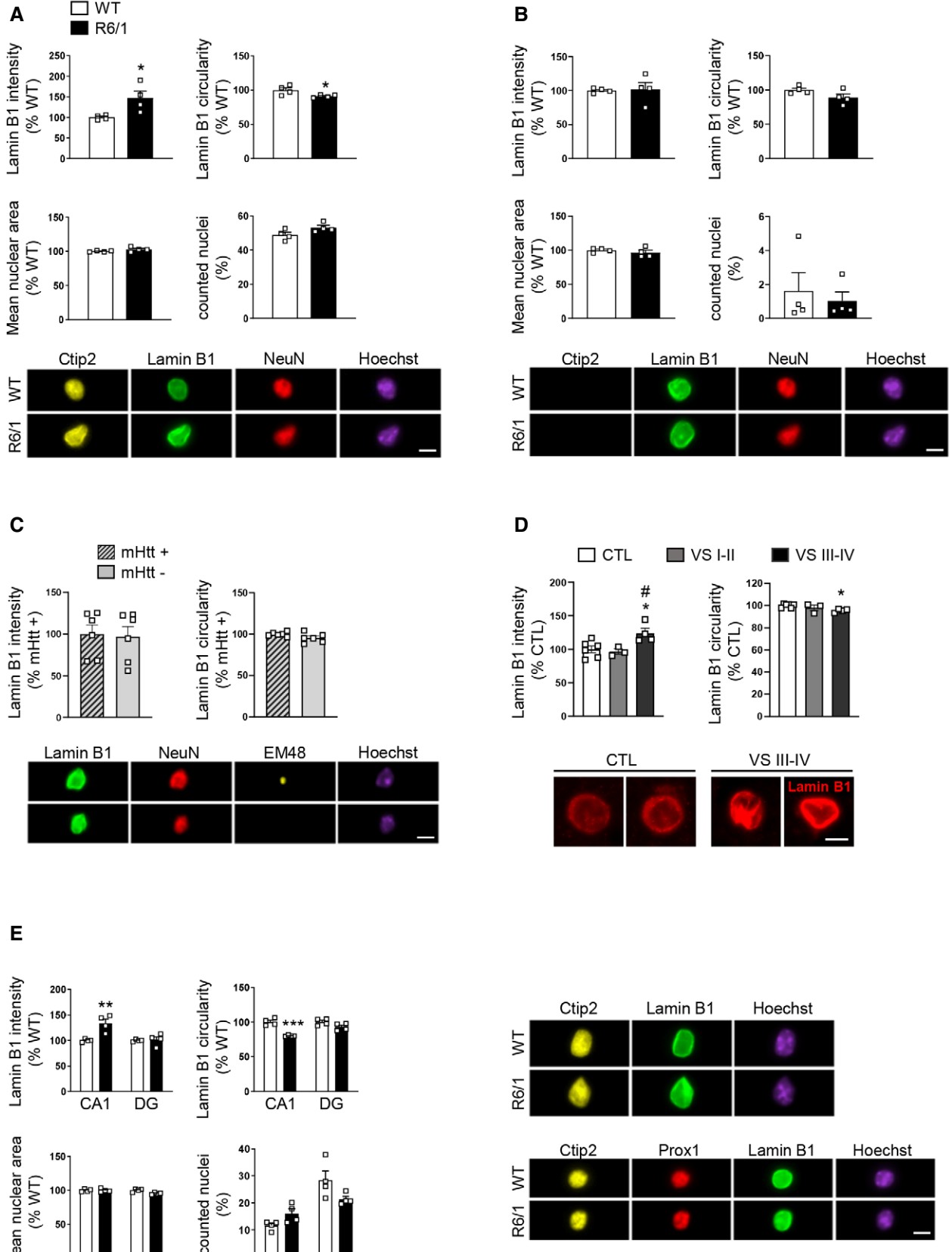

**Figure 4.**

**Figure 4. Increase in lamin B1 levels and altered nuclear morphology occur in striatal MSNs and hippocampal CA-1 neurons from R6/1 mice.**

Lamin B1 levels and morphology were analyzed in specific neuronal nuclei from the striatum and hippocampus of 30-week-old wild-type (WT) and R6/1 mice, and in the putamen of HD patients by FANSI and immunohistochemistry, respectively.

A  Graphs show the quantification of different parameters in mice striatal MSN nuclei (Ctip2+/NeuN+). $N = 4$ for both genotypes. An average of 5,000 nuclei were analyzed for each sample. Representative images are shown. Scale bar 7 μm.

B  Graphs show the quantification of different parameters in mice striatal interneurons (Ctip2-/NeuN+). $N = 4$ for both genotypes. An average of 5,000 nuclei were analyzed for each sample. Representative images are shown. Scale bar 7 μm.

C  Graphs show the quantification of lamin B1 intensity and circularity in 30-week-old R6/1 mouse striatal neuronal nuclei with (mHtt +) or without (mHtt −) nuclear inclusions. $N = 6$. EM48 antibody was used to label mHtt inclusions. Representative images are shown. Scale bar 7 μm.

D  Graphs showing lamin B1 intensity and circularity in MSN nuclei from the putamen of HD patients at different stages of the disease (VS: Vonsattel grade) and corresponding controls (CTL: non-affected individuals). $N = 6$ for CTL, $N = 3$ for VS I-II, and $N = 4$ for VS III-IV. An average of 50 nuclei were examined for each sample. Representative images (maximal Z-projections) are shown. Scale bar 7 μm.

E  Graphs show the quantification of different parameters in hippocampal CA1 (Ctip2+/Prox1−) and DG (Ctip2+/Prox1+) neuronal nuclei. $N = 4$ for each genotype. Representative images are shown. Scale bar 7 μm.

Data information: Each point corresponds to the value from an individual sample. Bars represent the mean ± SEM. Data in (A-C) and (E) were analyzed by two-tailed unpaired Student's *t*-test. *$P < 0.05$, **$P < 0.01$, and ***$P < 0.001$ as compared with corresponding controls. Data in (D) were analyzed by one-way ANOVA followed by Tukey's *post hoc* test. *$P < 0.05$ as compared with CTL; #$P < 0.05$ as compared with VS I-II. Exact $P$ values are reported in Appendix Table S3.

Source data are available online for this figure.

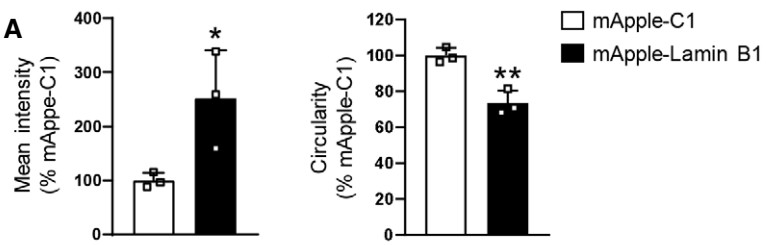

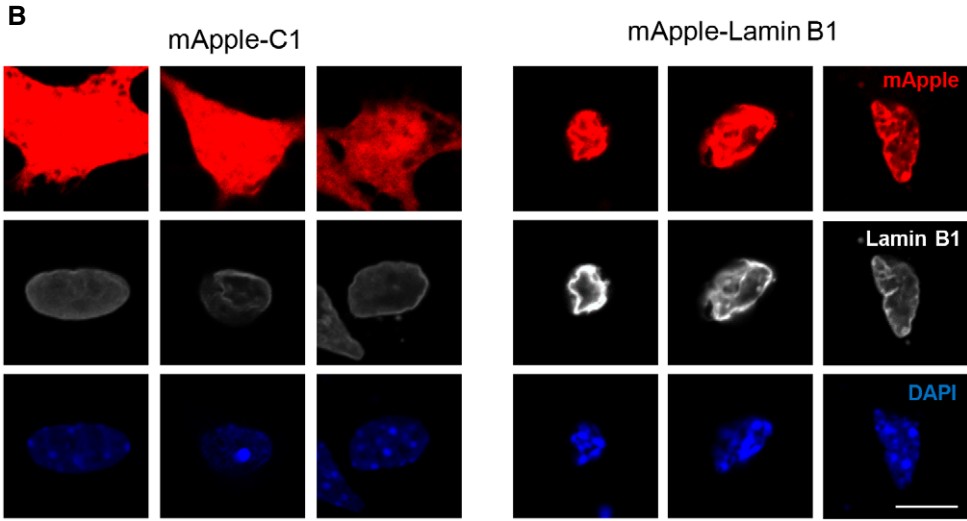

**Figure 5. Lamin B1 overexpression in primary striatal neurons alters nuclear morphology.**

Primary striatal neurons were transfected with a vector to overexpress lamin B1 (mApple-LB1) or with an empty vector (mApple-C1), and lamin B1 intensity and nuclear circularity were examined by immunocytochemistry against lamin B1 24 h after transfection.

A  Graphs show the quantification of lamin B1 intensity and circularity. $N = 3$. An average of 12 nuclei were examined in each culture. Data are expressed as a percentage of controls. Bars represent the mean ± SEM. *$P < 0.05$ and **$P < 0.01$ as compared to mApple-C1 control neurons (two-tailed unpaired Student's *t*-test). Exact $P$ values are reported in Appendix Table S3.

B  Representative images showing primary striatal neurons transfected with mApple-C1 or with mApple-Lamin B1, both in red. Neuronal nuclei were stained with DAPI Fluoromount-G (blue). Lamin B1 is shown in white. Scale bar 10 μm.

Source data are available online for this figure.

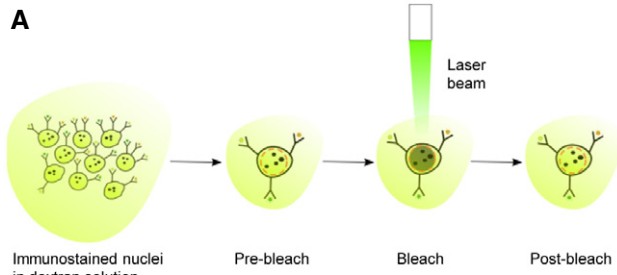

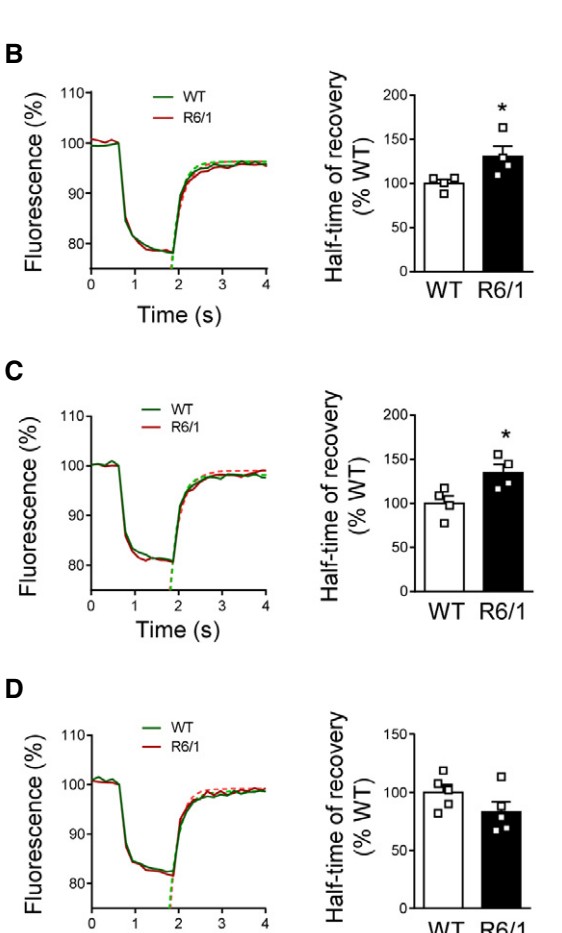

**Figure 6. Altered nuclear permeability in R6/1 mice striatal MSNs and hippocampal CA1 neuronal nuclei.**

A  Scheme showing the experimental approach followed to measure nuclear permeability by FRAP.

B–D  Nuclear permeability in 30-week-old wild-type (WT) and R6/1 mice striatal MSNs, CA1, and DG neuronal nuclei, respectively. (B and C) *N* = 4 for each condition; (D) *N* = 5 for each condition. An average of 25 nuclei were analyzed for each sample.

Data information: In graphs, each point corresponds to the value from an individual sample. Bars represent the mean ± SEM. *P < 0.05 as compared with WT mice (two-tailed unpaired Student's *t*-test). Exact *P* values are reported in Appendix Table S3.
Source data are available online for this figure.

accessibility in distal regulatory elements in R6/1 mice hippocampus. Overall, our results suggest that while no massive changes at transcriptional or chromatin accessibility levels are associated with

the loss of lamin B1 heterochromatin binding, a small subset of differentially expressed genes could be affected. Moreover, distal regulatory elements, and more particularly those found in regions with gained accessibility in R6/1 mice, appear to be the more sensitive to these lamin B1 alterations.

**Treatment with betulinic acid prevents cognitive impairment in R6/1 mice**

Given the increase in lamin B1 levels and the altered nuclear morphology and function found in R6/1 mice brain neurons, we hypothesized that these disturbances could be possibly contributing to motor and cognitive impairment present in HD. Betulinic acid has been shown to transcriptionally repress *LMNB1* expression (Li *et al*, 2013). Thus, as a clinical translational approach, we treated R6/1 mice from 8 to 20 weeks of age with 50 mg/kg betulinic acid and analyzed behavioral, biochemical, and histopathological changes following the timeline depicted in Fig 10A. First, we analyzed hippocampal-dependent learning and memory by using novel object location test (NOLT) and novel object recognition test (NORT). As previously described (Garcia-Forn *et al*, 2018), vehicle-treated R6/1 mice showed impaired hippocampal-dependent learning and memory relative to wild-type mice with a decreased percentage of time exploring the moved (Fig 10B) or the novel (Fig 10C) objects. Interestingly, betulinic acid-treated R6/1 mice explored similarly to wild-type mice in the NOLT (Fig 10B) and NORT (Fig 10C), suggesting that chronic administration of betulinic acid prevents cognitive dysfunction in R6/1 mice. To assess whether learning of a corticostriatal motor task was also improved after treatment with betulinic acid, we performed the accelerating rotarod task at 15 weeks of age, when R6/1 mice show a clear difference in the performance compared with wild-type mice (Garcia-Forn *et al*, 2018). Vehicle-treated R6/1 mice displayed poor performance with a lower latency to fall compared with control mice (Fig 10D). Importantly, betulinic acid-treated R6/1 mice showed a significant, although partial, improvement of their motor learning abilities.

Next, we investigated whether chronic betulinic acid treatment affected lamin B1 protein levels in different brain regions. A reduction in lamin B1 protein levels in the cortex and hippocampus, but not in the striatum, was detected in betulinic acid-treated compared with vehicle-treated R6/1 mice (Fig 10E). Analysis using FANSI revealed a normalization of lamin B1 levels in the nuclei of hippocampal CA1 neurons in betulinic acid-treated R6/1 mice (Fig 10F), accompanied by a partial rescue of nuclear morphology alterations (Fig 10F). In line with these results, we observed an amelioration of nucleocytoplasmic transport dysfunction, since half-time of recovery after photobleaching of nuclear dextran fluorescence was similar in betulinic acid-treated R6/1 and vehicle-treated wild-type mice hippocampal CA1 neuronal nuclei (Fig 10G). Importantly, as observed at 30 weeks of age, lamin B1 levels and nuclear morphology or permeability alterations were not detected in hippocampal DG neuronal nuclei from 20-week-old vehicle- or betulinic acid-treated R6/1 mice (Appendix Fig S12A and B).

Finally, we analyzed whether several hallmarks of the disease were affected by the treatment with betulinic acid. We found that betulinic acid did not prevent the loss in DARPP-32 levels in the striatum of R6/1 mice, whereas it completely prevented the loss in hippocampal PSD-95 levels (Appendix Fig S12C and D). Moreover,

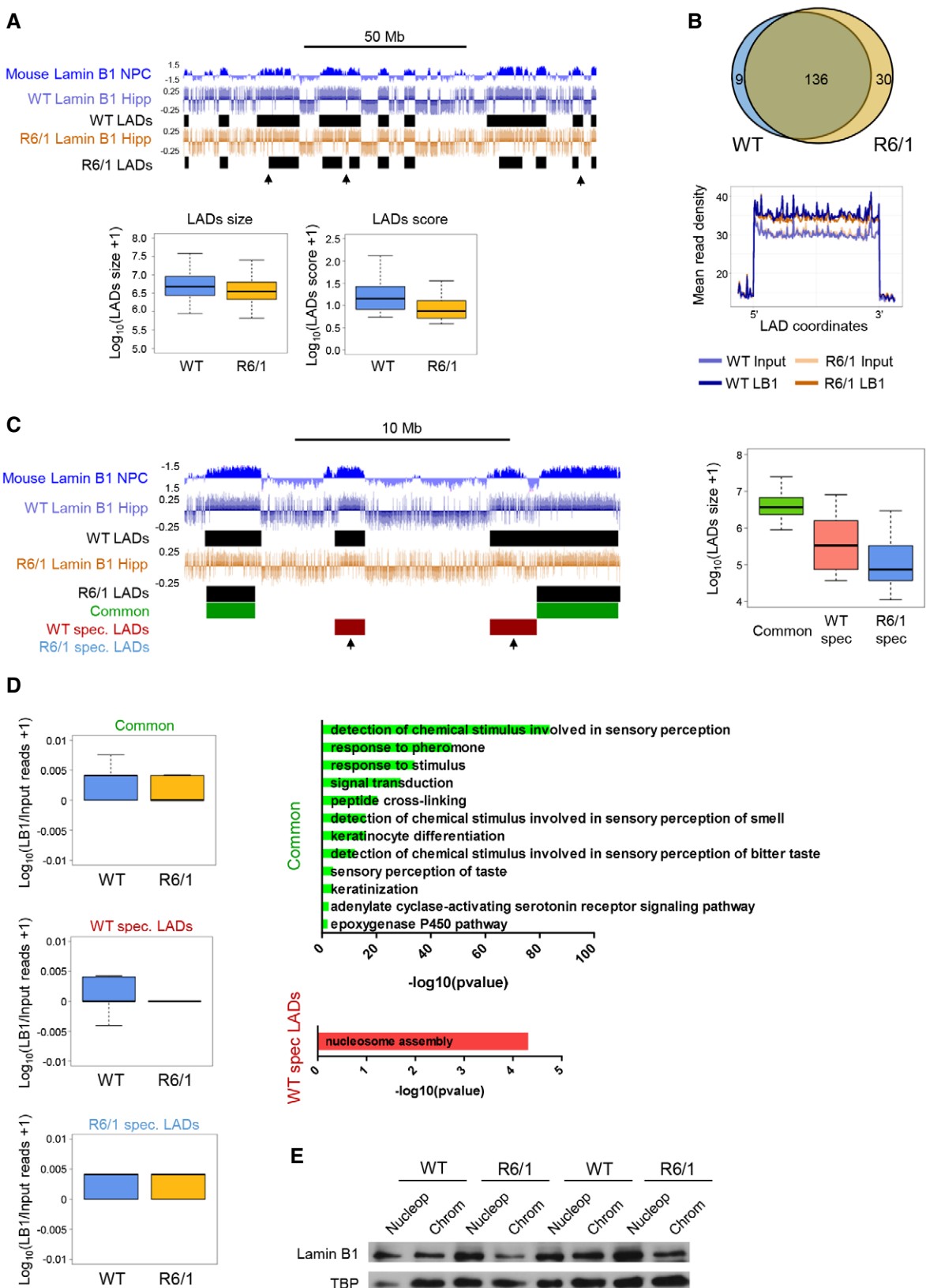

**Figure 7.**

**Figure 7. Lamin B1 chromatin binding in wild-type and R6/1 mice hippocampus.**

A   UCSC genome browser capture of lamin B1 ChIP-seq signal (log(LB1/Input)) and LADs discovered in wild-type (WT LAD) and R6/1 (R6/1 LADs) mice in combination with NPC lamin B1 DamID (top). Black arrows highlight differences in identified LADs between WT and R6/1 mice ChIP-seq data. Box plots of LAD size ($\log_{10}$(LAD size + 1)) and of LAD score ($\log_{10}$(LAD score + 1)) obtained by EDD (bottom) from WT ($N = 3$) and R6/1 ($N = 3$) lamin B1 ChIP-seq data. Hipp, hippocampus. The bottom and top of the boxes are the first and third quartiles, and the line within represents the median. The whiskers denote the interval within 1.5 times the interquartile range (IQR) from the median.

B   Venn diagram of overlapping LADs between wild-type (WT, $N = 3$) and R6/1 ($N = 3$) mice hippocampus (top). Metaprofile of lamin B1 and input datasets mean read density within common LADs for WT and R6/1 mice (bottom).

C   UCSC genome browser capture of lamin B1 ChIP-seq signal (log(LB1/Input)) LADs discovered in wild-type (WT LADs) and R6/1 (R6/1 LADs) mice; and common (green), wild-type (WT)-specific (WT spec. LADs, red), and R6/1-specific (R6/1 spec. LADs, blue) LADs identified by EDD (top, WT ($N = 3$), R6/1 ($N = 3$)). Black arrows highlight WT-specific LADs not identified in R6/1 mice. Box plot of average size ($\log_{10}$(LAD size + 1)) for common, WT-specific, and R6/1-specific LADs (right). The bottom and top of the boxes are the first and third quartiles, and the line within represents the median. The whiskers denote the interval within 1.5 times the interquartile range (IQR) from the median.

D   Lamin B1 enrichment in common, wild-type (WT, $N = 3$)-specific (spec), and R6/1 ($N = 3$)-specific LADs ($\log_{10}$(LB1/Input reads + 1) in hippocampus (Hipp; left). Bar graphs of significant (Benjamini's adjusted $P$-value < 0.05) Biological Processes terms from DAVID for genes within common and WT-specific LADs (right). Gene-term enrichment was estimated by DAVID using a modified Fisher's exact test and Benjamini's multiple correction test. Bars represent the $-\log_{10}$ (Benjamini's adjusted $P$-value). Exact $P$ values are reported in Appendix Table S3.

E   Representative immunoblot showing lamin B1 levels in the nucleoplasm (Nucleop) and chromatin (Chrom) in the hippocampus of 30-week-old wild-type (WT, $N = 7$) and R6/1 ($N = 7$) mice. TBP, TATA-binding protein.

Source data are available online for this figure.

we found that the number and area of mHtt aggregates in betulinic acid-treated R6/1 mice were similar to those found in the vehicle-treated group, both in the striatum and in the hippocampus (Appendix Fig S12E and F), indicating that betulinic acid effects were independent from mHtt aggregation.

## Discussion

Here, we show that (I) lamin B1 protein levels are increased in vulnerable regions of HD brain correlating with altered nuclear morphology; (II) nucleocytoplasmic transport of small molecules is altered in neurons showing increased lamin B1 levels in R6/1 mouse brain; (III) in R6/1 mice hippocampus (a) lamin B1 alterations correlate with partial unstructuring of LADs and (b) changes in chromatin accessibility mostly localize at distal regulatory elements, correlate with transcriptional dysregulation, and are partially associated with lamin B1 chromatin-binding alterations; and (IV) pharmacologic regulation of lamin B1 levels improves nucleocytoplasmic transport in CA1 hippocampal neurons and ameliorates behavioral abnormalities in R6/1 mouse.

We found that, among all lamin isoforms, lamin B1 was consistently affected in the brain of both HD patients and mouse models from early stages of the disease. B-type, but not A-type lamins, are essential for brain development (Kim *et al*, 2011). In fact, none of the laminopathies related to mutations in lamin A/C courses with neuronal dysfunction, which could explain why lamin A/C was unaltered in most R6/1 brain regions. Alterations in B-type lamins have been reported in neurodegenerative disorders and aging. For instance, and in contrast to our results, decreased lamin B levels are found in brains from AD (Frost *et al*, 2016) and PD patients bearing the LRRK2 (G2019S) mutation (Liu *et al*, 2012), and in aged primary human fibroblasts (Freund *et al*, 2012) and keratinocytes (Dreesen *et al*, 2013), being a marker of cellular senescence. Here, we show that increased lamin B1 levels in the brain of HD patients are not due to the aging process itself but rather to the pathogenic process. Interestingly, autosomal dominant leukodystrophy, a laminopathy caused by the duplication of the *LMNB1* gene, courses with severe central nervous system affectation, whose symptoms recall those of HD (Padiath *et al*, 2006). These evidences prompt the idea that increased lamin B1 levels in HD brain may be participating in the pathophysiology of the disease.

The accumulation of lamin B1 in neuronal nuclei from HD brain could be produced by different mechanisms such as increased transcription and/or translation, or decreased degradation. Increased transcription seems improbable since RNA-seq data generated from hippocampus of 30-week-old R6/1 mice do not show alterations in lamin B1 RNA levels (present results) (Hervás-Corpión *et al*, 2018). In addition, although we have recently showed increased translation in the striatum of HD mouse models and patients, the proteomic analysis did not reveal lamin B1 as one of the proteins with increased translation (Creus-Muncunill *et al*, 2019). Therefore, we hypothesized that increased lamin B1 protein levels in HD brain could be the result of different altered post-translational mechanisms such as decreased PKCδ (Rué *et al*, 2014) and/or altered autophagy-mediated lamin B1 degradation (Dou *et al*, 2015) and/or increased stabilization due to overactivation of p38MAPK (Barascu *et al*, 2012). Here, we show that knock-down of PKCδ in striatal cells increases lamin B1 levels in their nuclei, and that decreased PKCδ levels correlate with increased lamin B1 levels in the striatum, but not in the hippocampus, of R6/1 mice. Therefore, our results suggest that mechanisms leading to increased lamin B1 levels in HD brain could differ in a region-dependent manner, with downregulation of PKCδ playing a role in the striatum.

Immunohistochemical analysis of lamin B1 in R6/1 mouse brain suggested that the increase occurred in striatal MSNs, hippocampal CA1, and DG neurons. Our newly developed technique, FANSI, confirmed the increase in lamin B1 levels in nuclei of R6/1 striatal MSNs and showed that in the hippocampus CA1 neurons, but not DG neurons, were affected. Interestingly, these affected populations are the most vulnerable neurons in HD brain and their dysfunction participate in the motor and cognitive phenotype (Vonsattel & DiFiglia, 1998; Murphy *et al*, 2000). In the putamen of HD patients, FANSI results indicated a significant loss of MSNs, but not of glial cells, as previously described (Vonsattel *et al*, 1985). However, this technique seemed unsuitable to analyze nuclear alterations in MSNs

from HD patients as, after sample processing, the number of nuclei remaining was very low. In addition, when these nuclei were analyzed, they did not present differences in comparison with the nuclei from non-affected individuals. In contrast, the analysis of lamin B1 levels by immunohistochemistry in HD patient MSN nuclei revealed similar alterations to those seen in R6/1 mice MSNs.

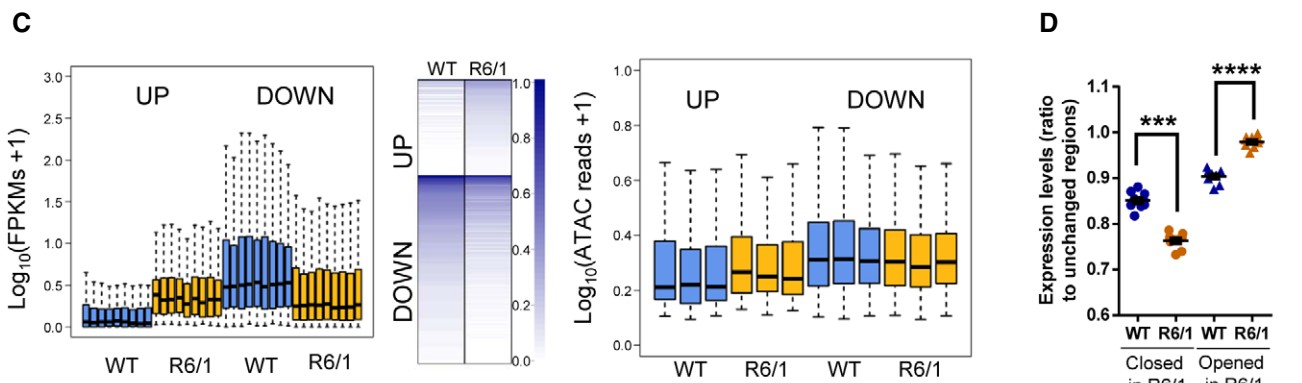

**Figure 8.**

**Figure 8. Hippocampal chromatin accessibility and gene expression analysis in R6/1 mice hippocampus.**

A   Scheme showing major steps of ATAC-seq technique.

B   UCSC genome browser capture of wild-type (WT, $N = 3$) and R6/1 ($N = 3$) mice hippocampus ATAC-seq data, hippocampal H3K9ac, and unchanged, closed in R6/1, and open in R6/1 accessible detected regions (left) in *Tuba1α*, *Gabra2α*, and *Dlx2* gene locus. Arrows in blue (closed in R6/1) and red (open in R6/1) indicate differential accessible regions. Bar graphs of significant (Benjamini's adjusted *P*-value < 0.05) Biological Processes terms from DAVID for genes associated with decreased (right top) or increased (right bottom) chromatin accessibility regions. Gene-term enrichment estimated by DAVID using a modified Fisher's exact test and Benjamini's multiple correction test. Bars represent the $-\log_{10}$ (Benjamini's adjusted *P*-value). Exact *P* values are reported in Appendix Table S3.

C   Box plots showing average gene expression ($\log_{10}$(FPKMs + 1) for genes up- or down-regulated (adjusted *P*-value < 0.001, IFCI>2) in R6/1 ($N = 9$) versus WT ($N = 9$) mice (left). The bottom and top of the boxes are the first and third quartiles, and the line within represents the median. The whiskers denote the interval within 1.5 times the interquartile range (IQR) from the median. Heat map showing expression profile ($\log_{10}$ (FPKMs + 1)) of genes up- or down-regulated (adjusted *P*-value < 0.001, IFCI>2, $N = 9$) in R6/1 mice (mid). Genes are ranked by the degree of expression. Numbers in color scale show the correspondence between gene expression values and colors. Box plots showing average TSS chromatin accessibility ($\log_{10}$(ATAC reads + 1), $N = 3$) for genes up- or down-regulated (adjusted *P*-value < 0.001, IFCI>2, $N = 9$) in R6/1 versus WT mice hippocampus (right). Exact *P* values are reported in Appendix Table S3. The bottom and top of the boxes are the first and third quartiles, and the line within represents the median. The whiskers denote the interval within 1.5 times the interquartile range (IQR) from the median.

D   Average gene expression (closed or open regions FPKMs/ unchanged regions FPKMs) for genes associated with differential accessible regions in R6/1 ($N = 9$) versus WT ($N = 9$) mice (adjusted *P*-value < 0.05, $N = 3$). Each point corresponds to the value from an individual sample. Data are shown as the mean ± SEM. *$P$ < 0.05 as compared with WT mice (two-tailed unpaired Student's *t*-test). Exact *P* values are reported in Appendix Table S3.

Source data are available online for this figure.

Therefore, we speculate that nuclear abnormalities potentiate MSN loss after processing of the tissue, probably due to repeated centrifugation, besides the neurodegenerative process itself. This could result in the evaluation of those MSNs with a healthier nuclear envelope (including "normal" lamin B1 levels) and not all the MSNs remaining in the tissue. Indeed, immunohistochemistry analyses confirmed an increase in lamin B1 levels in striatal neuronal nuclei from VS III-IV HD patients in comparison with control individuals. Altogether, our data support the idea of a cell-type-dependent increase in lamin B1 levels, specifically in those neurons preferentially affected in HD.

Our results show that increased lamin B1 levels correlate with altered nuclear morphology in the R6/1 mice brain-specific neurons. In fact, we show that the overexpression of lamin B1 in striatal cultured neurons alters nuclear morphology. In agreement with our results: (i) Nuclear morphology alterations in autosomal dominant leukodystrophy brain cells have been related to increased lamin B1 levels (Ferrera *et al*, 2014), and (ii) by using lamin B1 as a marker, nuclear envelope abnormalities have been shown in the brain of HD patients and mouse models (Gasset-Rosa *et al*, 2017). Furthermore, decreased levels of lamin B1 also result in nuclear morphology alterations in AD (Frost *et al*, 2016) and PD (Liu *et al*, 2012) neurons suggesting that proper lamin B1 levels are necessary to maintain a correct neuronal nucleus morphology. In addition, nucleocytoplasmic transport was altered in those nuclei with increased lamin B1 levels in accordance with previous literature showing a link between alterations in lamins and nuclear dysfunction. For instance, in Hutchinson–Gilford progeria syndrome, the mutated form of A-type lamin induces nucleocytoplasmic transport dysregulation by inhibiting the nuclear localization of Ubc9 and disrupting the nucleocytoplasmic Ran gradient, necessary for active transport (Kelley *et al*, 2011). In addition, cells expressing progerin presented perturbed passive and active transport toward and from the nucleus (Ferri *et al*, 2017). Similarly, increased levels of lamin B1 affect

**Figure 9. Chromatin accessibility, gene expression, and lamin B1 chromatin-binding interconnexion.**

A   Box plots of average expression ($\log_{10}$(FPKMs + 1)) for genes sublists 1-5 (lowest to highest expression) for wild-type (WT, $N = 3$) mice (left). Pie charts of gene distribution among generated sublists (1-5) for all genes (middle) and genes in LADs (right) in WT mice. The bottom and top of the boxes are the first and third quartiles, and the line within represents the median. The whiskers denote the interval within 1.5 times the interquartile range (IQR) from the median.

B   Average gene expression (FPKMs) of genes found in common and wild-type (WT, $N = 3$)-specific LADs for WT ($N = 9$) and R6/1 ($N = 9$) mice. Each point corresponds to the value from an individual sample. Data are shown as the mean ± SEM. * $P$ < 0.05 as compared to WT mice (two-tailed unpaired Student's *t*-test). Exact *P* values are reported in Appendix Table S3.

C   Average chromatin accessibility (normalized read counts) of genes found in common and wild-type (WT, $N = 3$)-specific LADs for WT ($N = 3$) and R6/1 ($N = 3$) mice. Each point corresponds to the value from an individual sample. Data are shown as the mean ± SEM. The Wilcoxon–Mann–Whitney test was used for statistical analysis. Exact *P* values are reported in Appendix Table S3.

D   Venn diagram showing the total number of genes found in common and wild-type (WT)-specific LAD regions (left). Bar graph showing the number of up- and down-regulated genes found in common and WT-specific LAD regions filtering only according to adjusted *P*-value (adjusted *P*-value < 0.001) or additionally with fold change (adjusted *P*-value < 0.001, IFCI > 2) obtained from deseq2 differential expression analysis (see methods).

E   UCSC genome browser capture of wild-type (WT, $N = 3$) and R6/1 ($N = 3$) mice hippocampus ATAC-seq data, lamin B1 ChIP-seq ($\log$(Lb1 ChIP/Input)) for WT ($N = 3$) and R6/1 ($N = 3$) mice hippocampus, hippocampal H3K9ac, CTCF, and H3K9me3 for WT mice hippocampus in *Fos* locus (left). Enhancer regions (E1-E5) are indicated with arrows. Box plot of lamin B1 enrichment ($\log_{10}$(Lb1 ChIP/Input)) for regions with unchanged, decreased (closed in R6/1), and increased (opened in R6/1) chromatin accessibility in R6/1 ($N = 3$) mice hippocampus. * $P$ < 0.05 as compared to WT ($N = 3$) mice (the Wilcoxon–Mann–Whitney test). Exact *P* values are reported in Appendix Table S3. The bottom and top of the boxes are the first and third quartiles, and the line within represents the median. The whiskers denote the interval within 1.5 times the interquartile range (IQR) from the median.

Data information: In all graphs, bars represent the mean ± SEM and each point corresponds to the value from an individual sample. Statistical analysis was performed by one-way ANOVA followed by Bonferroni's post hoc test except in (D) where data were analyzed by two-way ANOVA followed by Bonferroni's *post hoc* test. Exact *P* values are reported in Appendix Table S3.

Source data are available online for this figure.

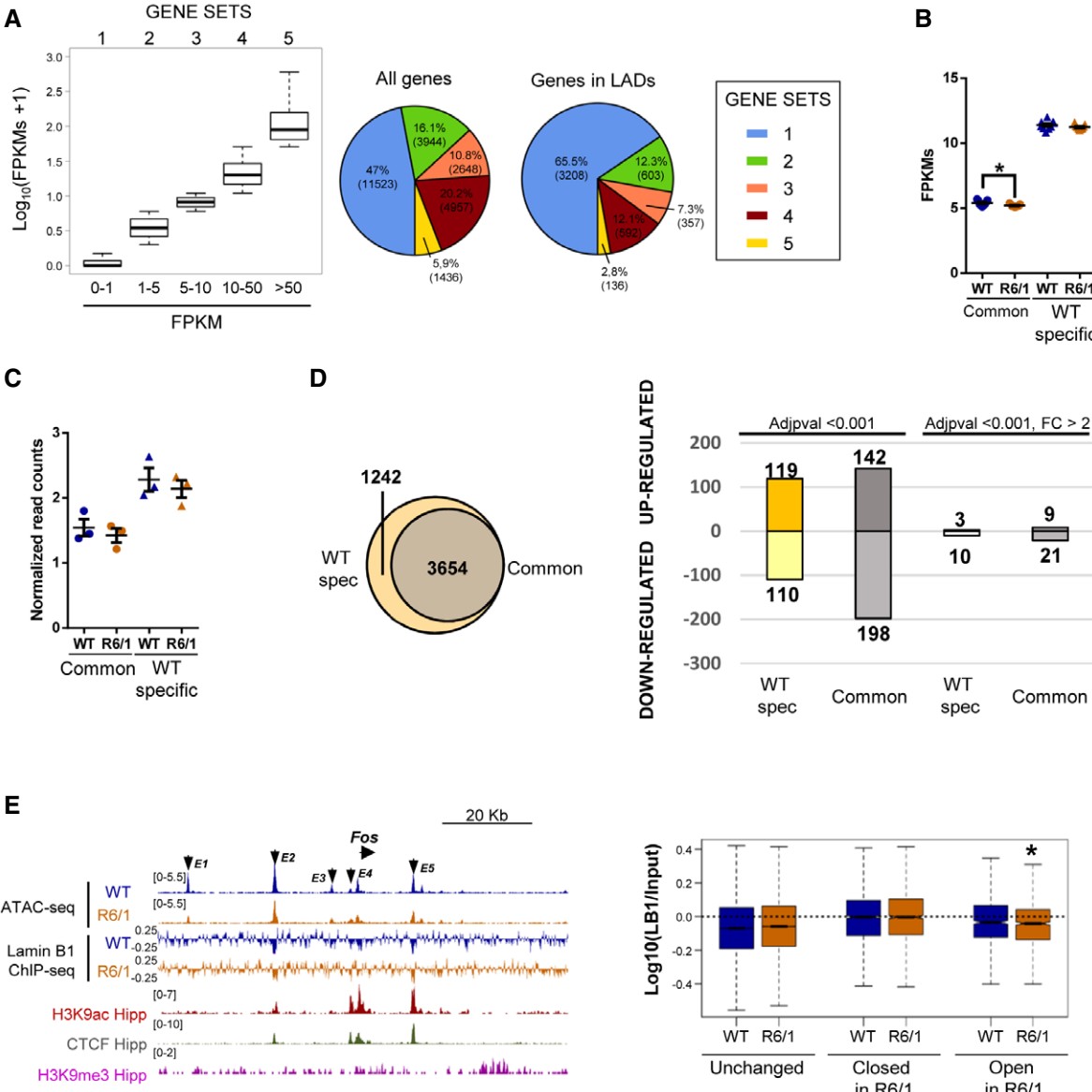

**Figure 9.**

nuclear export in HEK293 cells (Lin & Fu, 2009) and reduces nuclear ion channel opening in fibroblasts from autosomal dominant leukodystrophy patients (Ferrera *et al*, 2014). Furthermore, altered lamin B levels and nucleocytoplasmic transport occur in AD (Eftekharzadeh *et al*, 2018) and PD (Chalovich *et al*, 2006) although these alterations have never been linked between them. Altered nucleocytoplasmic mRNA transport has been previously reported in other HD models and has been indirectly associated with nucleoporins sequestration by mHtt inclusions (Gasset-Rosa *et al*, 2017; Grima *et al*, 2017). In contrast, our immunohistochemical and FANSI analyses indicated that increased lamin B1 and morphological alterations occur in R6/1 mouse striatal and hippocampal neuronal nuclei with and without the presence of mHtt inclusions. These differences may be due to the use of different HD models, in which forms of mHtt aggregation differ (Morton *et al*, 2000; Hansson *et al*, 2001; Heng *et al*, 2010). Altogether, here we show

nuclear morphology and nucleocytoplasmic transport abnormalities in HD brain that occur in a neuron type-dependent manner and that are related, at least in part, to increased lamin B1 protein levels.

In order to study the consequences of lamin B1 alterations in nuclear lamina heterochromatin organization, we generated, for the first time, lamin B1 ChIP-seq data in mouse central nervous system tissue and characterized hippocampal LADs which, as expected, showed high homology with previous identified domains using DamID (Peric-Hupkes *et al*, 2010). Our lamin B1 ChIP-seq data, together with nuclear fractionation experiments, clearly showed a perturbation in nuclear lamina heterochromatin organization and lamin B1 chromatin binding. Interestingly, previous studies demonstrated that lamin B1 overexpression in the central nervous system leads to epigenetic alterations affecting the heterochromatin protein 1 β (HP1β) and methylated histone H3 (H3K9) as well as transcriptional programs mostly linked to glial cells (Lin & Fu, 2009). In line with

that, the striatum of the R6/2 mouse model of HD shows important alterations in H3K9me3 coverage (Lee *et al*, 2017) that, as we have shown, is highly enriched in LADs. While the high extent of transcriptional and chromatin accessibility alterations we have identified in R6/1 mice hippocampus can be hardly explained exclusively by the subtle

alterations identified in lamin B1 ChIP-seq data, our results indicated that regions with gained chromatin accessibility in our ATAC-seq data showed a global decrease in the binding of lamin B1, correlative increased expression, and were enriched in terms associated with cell division and development, more typically associated with glial than to

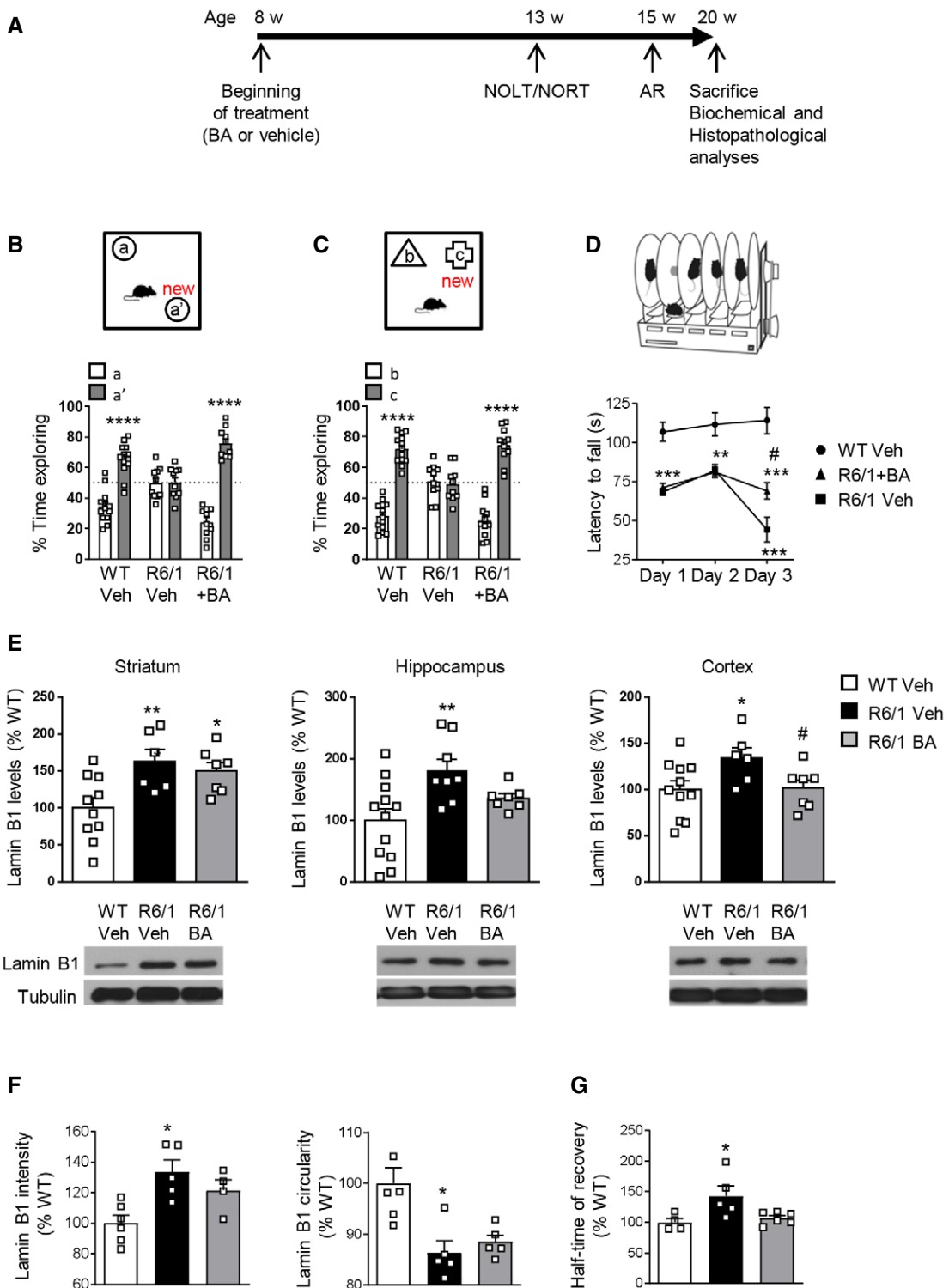

**Figure 10.**

**Figure 10. Chronic treatment with betulinic acid improves cognitive function and modulates lamin B1 levels in the hippocampus and cortex of R6/1 mice.**

A    Timeline of the behavioral, biochemical, and histopathological analyses performed in wild-type (WT) and R6/1 mice to assess the effect of betulinic acid (BA) administration. w, weeks; AR, accelerating rotarod.

B, C    Graphs show the percentage of time exploring each object with respect to the total exploration time in the (B) NOLT and (C) NORT, 5 weeks after treatment (Veh, vehicle; BA, betulinic acid; WT, wild-type). ****$P < 0.0001$ compared with the corresponding old location/object. WT veh $N = 13$; R6/1 veh $N = 10$; R6/1 + BA $N = 10$.

D    Accelerating rotarod was assessed after 7 weeks of treatment. **$P < 0.01$ and ***$P < 0.001$ compared with vehicle-treated wild-type (WT) mice; #$P < 0.05$ compared with vehicle-treated R6/1 mice. WT veh $N = 13$; R6/1 veh $N = 9$; R6/1 + BA $N = 9$.

E    Lamin B1 levels were analyzed by Western blot in the striatum (WT veh $N = 10$; R6/1 veh $N = 6$; R6/1 + BA $N = 7$), hippocampus (WT veh $N = 12$; R6/1 veh $N = 8$; R6/1 + BA $N = 7$), and cortex (WT veh $N = 11$; R6/1 veh $N = 6$; R6/1 + BA $N = 7$) after 12 weeks of treatment. *$P < 0.05$, **$P < 0.05$ compared with vehicle-treated wild-type (WT) mice and #$P < 0.05$ compared with vehicle-treated R6/1 mice. Representative immunoblots of lamin B1 and α-tubulin (as loading control) for each treatment group are shown.

F    Lamin B1 intensity and nuclear morphology were analyzed by FANSI in hippocampal CA1 neuronal nuclei from wild-type (WT) and R6/1 mice after 12 weeks of treatment. *$P < 0.05$ compared with vehicle-treated WT mice. *$P < 0.05$ as compared to vehicle-treated WT mice. Lamin B1 intensity: WT veh $N = 6$; R6/1 veh $N = 5$; R6/1 + BA $N = 4$; lamin B1 circularity: WT veh $N = 6$; R6/1 veh $N = 5$; R6/1 + BA $N = 5$.

G    Nuclear permeability was analyzed by FRAP in hippocampal CA1 neuronal nuclei after 12 weeks of treatment. *$P < 0.05$ compared with vehicle-treated wild-type (WT) mice. WT veh $N = 4$; R6/1 veh $N = 5$; R6/1 + BA $N = 6$.

Data information: In all graphs, bars represent the mean ± SEM and each point corresponds to the value from an individual sample. Statistical analysis was performed by one-way ANOVA followed by Bonferroni's post hoc test except in (D) where data were analyzed by two-way ANOVA followed by Bonferroni's *post hoc* test. Exact *P* values are reported in Appendix Table S3.

Source data are available online for this figure.

neuronal cells (Merienne *et al*, 2019), suggesting that chromatin relaxation and consequent induction of gene expression could be, at least in part, a result of the loss of lamin B1 chromatin binding. In accordance with this, a previous study showed a general gain of chromatin accessibility in HD T cells (Park *et al*, 2017), while our hippocampal ATAC-seq data showed a bi-directionality in chromatin accessibility changes, with a clear compaction of neuronal-associated regulatory regions and increased chromatin relaxation in developmental related ones. This is in agreement with a recent study demonstrating that HD neuronal and glial cells are affected in opposite ways at transcriptional level (Merienne *et al*, 2019). Our R6/1 hippocampal transcriptional data showed a great overlap with previously generated data in additional HD models (Valor *et al*, 2013; Langfelder *et al*, 2016), and according to a recent study, these common transcriptional signatures present high homology with those found in knockouts for histone acetyltransferases and methyltransferases (Hervás-Corpión *et al*, 2018). Being shown the interplay between lamin B1 protein levels and H3K9me3, highly dependent on the activity of methyltransferases, it can be speculated that nuclear lamina alterations identified in the present work could lead to a reorganization of H3K9me3 coverage in HD hippocampus. Altogether, our data suggest a new mechanism contributing to the complex epigenetic landscape of HD (reviewed in Francelle *et al*, 2017).

Finally, with the purpose of addressing the therapeutic relevance of our findings, we used betulinic acid, which has the potential to normalize lamin B1 protein levels (Li *et al*, 2013). We show a beneficial effect in preventing HD cognitive dysfunction and, for the first time, in normalizing lamin B1 protein levels in the brain *in vivo*. Interestingly, we observed a functional recovery of hippocampal memory-dependent tasks, which at the molecular level coincides with a normalization of lamin B1 levels in the hippocampus of R6/1 mice, an improvement in nuclear morphology and function of CA1 neurons, and no effect on mHtt aggregates. The fact that nuclear improvement occurred despite the presence of mHtt aggregates supports a role of lamin B1 alterations in the nuclear dysfunction previously observed in HD (Gasset-Rosa *et al*, 2017; Grima *et al*, 2017). In line with our results, cytotoxicity is reduced in primary

cortical neurons expressing mHtt after pharmacological restoration of nucleocytoplasmic transport (Grima *et al*, 2017). Our results strengthen the idea of a relationship between increased lamin B1 levels and alterations in nuclear morphology and function in HD, at least in CA1 hippocampal neurons, as previously suggested in autosomal dominant leukodystrophy (Ferrera *et al*, 2014), which ultimately contribute to the HD phenotype. In contrast, lamin B1 levels were not normalized in the striatum after betulinic acid treatment although partial amelioration of motor learning dysfunction was observed in R6/1 mice. Since cortical pyramidal neuronal project to the striatum and betulinic acid normalized lamin B1 levels in the cortex, we speculate that the improvement of cortical neuronal function could have beneficial effects on MSNs, reflected by the prevention in the decrease in DARPP-32 protein levels, a hallmark of HD (Bibb *et al*, 2000), and consequently, motor performance is partially improved. In addition to the regulation of lamin B1 levels, betulinic acid has been shown to exert beneficial effects in affected brain through the regulation of cAMP, cGMP, and BDNF levels (Kaundal *et al*, 2018), long-term potentiation (Navabi *et al*, 2018), oxidative stress (Lu *et al*, 2011), or inflammation (Li *et al*, 2018), which may contribute to the improvement of R6/1 mice cognitive behavior. Therefore, our results open a new therapeutic window not only for HD, but also for autosomal dominant leukodystrophy, for which no effective treatment is available yet (Padiath, 2019).

Altogether, our findings suggest a relationship between increased lamin B1 levels and nuclear morphological and functional alterations in specific HD brain neurons, which may contribute to the pathophysiology of the disease and could have promising applications at the therapeutic level.

# Materials and Methods

### HD mouse model

Male R6/1 transgenic mice (B6CBA background) expressing the exon 1 of mHtt with 145 CAG repeats and their wild-type littermate

controls were used for this study. Mouse genotyping and CAG repeat length determination were performed as previously described (Mangiarini et al, 1996). All mice were housed together in numerical birth order in groups of mixed genotypes, and male littermates were randomly assigned to experimental groups. Data were recorded for analysis by microchip mouse number, and experiments were conducted in a blind-coded manner respect to genotype. The animals were housed with access to food and water ad libitum in a colony room kept at 19–22°C and 40–60% humidity, under a 12:12-h light/dark cycle. All procedures were carried out in accordance with the National Institutes of Health Guide for the Care and Use of Laboratory Animals, and approved by the local animal care committee of the Universitat de Barcelona, following European (2010/63/UE) and Spanish (RD53/2013) regulations for the care and use of laboratory animals.

## Post-mortem human brain tissue

Frozen samples (putamen, hippocampus, and frontal cortex) and brain slices (5-μm-thick sections paraffin-embedded mounted in glass slides) from HD patients and control individuals were obtained from the Neurological Tissue Bank of the Biobank-Hospital Clínic-Institut d'Investigacions Biomèdiques August Pi i Sunyer (IDIBAPS; Barcelona, Catalonia, Spain) following the guidelines and approval of the local ethics committee (Hospital Clínic of Barcelona's Clinical Research Ethics Committee). Informed consent was obtained from all subjects, and experiments were performed conformed the principles set out in the WMA Declaration of Helsinki and Department of Health and Human Services Belmont Report. Details on the sex, age, CAG repeat length, Vonsattel grade, and post-mortem delay are found in Appendix Table S1.

## Pharmacological treatment

R6/1 mice were treated (from 8 to 20 weeks of age) with vehicle (90% water, 10% polysorbate 80) or betulinic acid (50 mg/kg; Sigma-Aldrich, #855057) administered by oral gavage, 3 days/week. Wild-type mice were treated with vehicle. Animal weight was recorded each day of treatment. Days in which treatment and tests were coincident, mice were allowed to recover during 1 h before starting a task. Mice were sacrificed by cervical dislocation 1 h after the last dose. Half of the brain was fixed in 4% PFA for immunostaining analysis, and the striatum, hippocampus, and cortex from the other half were rapidly removed and stored at –80°C until analysis.

## Behavioral assessment

### Spatial and recognition memory tests
NOLT and NORT were used to analyze hippocampal-dependent spatial long term and recognition memory, respectively, in wild-type and R6/1 mice at 13 weeks of age as previously described (Garcia-Forn et al, 2018). In each of the tests, the object preference was measured as the time exploring each object × 100/time exploring both objects. The arena and the objects were rigorously cleaned between animal trials to avoid odors. Animals were tracked with SMART Junior software from Panlab (Barcelona, Spain). Days, in which treatment and tests were coincident, the mice were allowed to recover for 1 h after the treatment before starting any task.

### Accelerating rotarod
For the assessment of motor learning dependent on the corticostriatal connectivity, we performed the accelerating rotarod test at 15 weeks of age. The protocol was performed as previously described (Garcia-Forn et al, 2018) The final performance was calculated as the mean latency to fall during the 3 last trials of each day. Days, in which treatment and tests were coincident, the mice were allowed to recover for 1 h after the treatment before starting any task.

## Knock-in striatal cell line

Conditionally immortalized wild-type huntingtin knock-in striatal cells, STHdh$^{Q7/Q7}$, were grown as previously described (Trettel et al, 2000) on 12-mm round glass coverslips and were transfected at 50% of confluence. PKCδ siRNA (ON-TARGETplus Mouse Prkcd siRNA-SMART Pool, Dharmacon) was used for PKCδ silencing, and a scramble siRNA (Silencer® Select Negative Control No. 1 siRNA, Catalog Number 4390844) was used as a control. To overexpress mHtt, an exon 1 Htt plasmid expressing 94 CAG repeats tagged with the CFP (Ortega et al, 2010) was used. Exon 1 mHtt plasmid (0.5 μg) was transfected at the same time as the siRNAs (10 pmol) using Lipofectamine 3000 (Invitrogen, Carlsbad, CA, USA), and cells were incubated for 24 h.

## Striatal primary cultures

Striatal primary cultures were obtained and maintained as previously described (Gratacòs et al, 2001) and grown on 12-mm round glass coverslips. At 9 days in vitro, they were transfected with a plasmid containing lamin B1 (mApple-lamin B1-10) or with a plasmid containing mApple-C1 as control. Both plasmids were a gift from Michael Davidson (Addgene plasmid, #54917 and #54631, respectively). Transfection was performed in 24-well plates using 2 μg of DNA and 1:2 TransFectin™ Lipid Reagent (Bio-Rad) following the manufacturer's instructions. After 45 min, medium was exchanged for 50% of fresh medium and 50% of neuronal conditioned medium. Cells were fixed 24 h post-transfection.

## Protein extraction and Western blot analyses

Animals were killed at different ages by cervical dislocation. Brains were quickly removed, and the striata, hippocampi, and cortex were dissected out and homogenized in lysis buffer. Protein extraction from mouse and human brain tissue, and from cultured cells, and Western blot analyses were performed as previously described (Xifró et al, 2008; Saavedra et al, 2011). After incubation with primary and the appropriated horseradish peroxidase-conjugated secondary antibodies (Appendix Table S2), membranes were washed with Tris-buffered saline containing 0.1% Tween 20. Immunoreactive bands were finally visualized using the Western Blotting Luminol Reagent (Santa Cruz Biotechnology, #sc-2048) and quantified by a computer-assisted densitometer (Gel-Pro Analyzer, version 4, Media Cybernetics).

## Immunofluorescence

Mice perfusion, brain processing, and immunostaining were performed as previously described (Rué et al, 2014). For human

tissue, the first step was dewaxing and rehydrating the tissue by performing a series of 5 min each: xylene (four times), absolute ethanol (three times), alcohol 96% (three times), and distilled water. The antigen retrieval was performed afterward by boiling the sections in citrate buffer (10 mM sodium citrate, 0.05% Tween 20, pH 6.0) in a microwave for 20 min. After this, the Dako Autostainer Plus was used for a blocking step during 15 min at room temperature with a commercial wash buffer from Dako supplemented with 3% normal goat serum, three washes with phosphate-buffered saline (PBS), and the incubation with the primary antibody in the Dako Real TM antibody diluent (Agilent, #S202230-2) for 30 min. After incubation with primary antibodies, sections were washed with PBS and incubated overnight with corresponding secondary antibodies (Appendix Table S2). Finally, sections were mounted with DAPI Fluoromount-G (Thermo Fisher Scientific, #00-4959-52). Negative controls were performed for each primary antibody, and no signal was detected in this condition.

For immunocytochemistry, cells were washed with PBS and fixed with 4% paraformaldehyde (PFA) in PBS for 10 min at room temperature. To block the action of PFA, cells were incubated with 0.2 M glycine for 20 min at room temperature. After quenching with 50mM $NH_4Cl$ for 10 min, cells were permeabilized in blocking buffer containing 1% BSA + 0.2% gelatin + 0.2% Triton X-100 in PBS at room temperature. After blocking, cells were incubated with rabbit anti-Lamin B1 (1:200; Abcam, Cambridge, UK) for 30 min at room temperature. Next, cells were washed three consecutive times with PBS and finally were incubated with Cy3 AffiniPure F(ab')2 Fragment Goat Anti-Rabbit IgG, F(ab')2 Fragment Specific (1:200, Jackson ImmunoResearch, West Grove, PA, USA). Nuclei were stained with DAPI Fluoromount.

## Immunofluorescence imaging and analysis

Immunostained tissue sections and STHdh$^{Q7/Q7}$ cells were examined by using the Olympus BX60 (Olympus, Tokyo, Japan) epifluorescence microscope coupled to an Orca-ER cooled CCD camera (Hamamatsu Photonics, Hamamatsu, Japan) or the Leica TCS SP5 laser scanning confocal microscope (Leica Microsystems Heidelberg GmbH, Manheim, Germany) with Argon and HeNe lasers coupled to a Leica DMI6000 inverted microscope at different magnifications (from 10× to 63×). Striatal cultured neurons, striatal olig-2-positive cells, and putamen z-stacks images were obtained with a Zeiss LSM 880 (Carl Zeiss Microscopy, LLC, Thornwood, NY, USA) confocal microscope using the ZEN acquisition software. Confocal images were taken as stacks differed in 0.5 μm (for mouse brain tissue), 0.3 μm (for human brain tissue), or 0.6 μm (for striatal primary neurons) in Z-axis with an HCX PL APO lambda blue 63× numerical aperture objective and standard pinhole (1 Airy disk), and their reconstruction was performed using ImageJ software (NIH, Bethesda, USA).

For morphological analysis in tissue sections, a Z-projection of confocal stack images was generated and nuclear segmentation was performed using default parameters of StarDist ImageJ plugin (Schmidt *et al*, 2018). A minimum of 400 (for mouse striatal sections) or 40 (for human putamen sections) nuclei were detected and computed for posterior morphological analysis. The morphological parameters of resulting segmented nuclei were analyzed using MorphoLibJ ImageJ plugin (Legland *et al*, 2016). Cultured cell nuclei

were analyzed using the ROI manager from ImageJ (an average of 20 and 12 nuclei per condition and culture were analyzed for STHdh$^{Q7/Q7}$ cells and striatal primary neurons, respectively).

## Immunohistochemistry for mHtt aggregates detection

Coronal sections (30 μm) of the whole brain were obtained as described above. Detection of mHtt aggregates was performed as previously described (Garcia-Forn *et al*, 2018) by using the anti-EM48 antibody. EM48 staining was examined in eight slices per animal separated by 240 μm (covering the entire striatum or CA1 hippocampal region) by using the Computer-Assisted Stereology Toolbox (CAST) software (Olympus Danmark A/S, Ballerup, Denmark). Images were analyzed using CellProfiler Analyst software.

## FANSI

FANSI was the result of combining nuclear purification and immunostaining (Benito *et al*, 2015) with ImageStream imaging flow cytometer technology (Barteneva & Vorobjev, 2016) (Luminex Corporation).

### Nuclear purification and immunostaining

Frozen tissue was homogenized in low sucrose buffer (LSB; 0.32 M sucrose, 5 mM $CaCl_2$, 5 mM $Mg(Ac)_2$, 0.1 mM EDTA, 50 mM HEPES pH 8.0, 1 mM DTT, 0.1% Triton X-100) and fixed in 1% formaldehyde for 10 min at room temperature in a rotating wheel. Formaldehyde was quenched with 125 mM glycine incubation during 5 min at room temperature in the rotating wheel. Tissue homogenate was collected by centrifugation, resuspended in LSB, and mechanically homogenized. After that, the homogenized solution was layered on the top of a high sucrose buffer (1 M sucrose, 3 mM $Mg(Ac)_2$, 10 mM HEPES, pH 8.0, 1 mM DTT) and centrifuged at 4°C to recover the nuclei. These were then resuspended in PBTB (PBS, 5% BSA, 0.1% Tween-20) containing the antibodies and 3% NHS, and incubated in a rotating wheel at 4°C during 30 min. After that, samples were washed twice with PBTB plus 3% NHS and stained with corresponding secondary antibodies in PBTB plus 3% NHS at 4°C during 15 min. Nuclei were then washed, stained with Hoechst 33258 (1:10,000; Thermo Fisher Scientific, #H3569), and directly processed for Imaging flow cytometry.

### Imaging flow cytometry (ImageStream)

Purified nuclei were resuspended in 100 μl and filtered using cells strainers of 50 μm pore size (Sysmex Partec, Kobe, Japan) and posteriorly sorted and imaged using a 60× objective at a maximum speed of 600 nuclei/s depending on the sample concentration. For each replicate, a minimum of 10,000 nuclei were recorded. Fluorescent minus one controls were used to evaluate the specificity of the defined populations by individually removing primary but not secondary antibodies (Appendix Fig S13A–F). Recorded files were processed and analyzed by the IDEAS Software provided by the ImageStream machine's manufacturers (Luminex, Austin, USA) after cross-channel signal compensation. After selecting individual nuclei (singlets, only focused acquired images were used for posterior analysis) (Appendix Fig S13F and G), lamin B1-positive nuclei were selected for posterior analysis (Appendix Fig S7I) and screened

according to their Ctip2 and Prox1 signal and the specific neuronal nuclear marker NeuN. For hippocampal samples, CA1 nuclei were classified as Ctip2$^+$/Prox1$^-$ and DG ones as Ctip2$^+$/Prox1$^+$ (Bagri et al, 2002; Arlotta et al, 2005). For striatum and putamen, nuclei were classified as MSNs (NeuN$^+$/Ctip2$^+$) (Herculano-Houzel & Lent, 2005; Arlotta et al, 2008), interneurons (NeuN$^+$/Ctip2$^-$), or glia (NeuN$^-$/Ctip2$^-$). All the features analyzed (mean intensity, circularity, and mean area) were performed using these selected populations.

### FRAP in isolated nuclei

Striatal nuclei were isolated as previously described (Alvarez-Periel et al, 2018) and then incubated for 30 min at 4°C with the corresponding primary and secondary antibodies (Appendix Table S2). Nuclei were maintained in LSB until analysis. To perform conventional single-photon FRAP experiments, nuclei were incubated with a solution of 20 kDa FITC-dextran (0.3mg/ml; Sigma-Aldrich, #FD20) and seeded in glass-bottomed chambers and covered with a cover slip. Striatal MSN nuclei (Ctip2$^+$/NeuN$^+$) and hippocampal CA1 (Ctip2$^+$/Prox1$^-$) and DG (Ctip2$^+$/Prox1$^+$) nuclei were manually selected. Each FRAP experiment started with 5 pre-bleach image scans, followed by 8 bleach pulses of 156 ms each on a spot with a diameter of 2.5 μm in the center of the nucleus. At the post-bleach period, a series of 100 single section images were collected at 156-ms intervals (Appendix Fig S14). A total of 113 images were acquired for each nucleus, and an average of 25 nuclei were analyzed for each animal. Image size was $256 \times 56$ pixels, and the pixel width was $120 \times 60$ nm. For imaging, the laser power was attenuated to 3% of the bleach intensity. FRAP experiments were performed on a Leica TCS SP5 laser scanning confocal spectral microscope (Leica Microsystems, Heidelberg, Germany) equipped with Argon laser and Leica DMI6000 inverted microscope. Images were acquired using a 63×, 1.4 NA oil immersion objective lens, and 1.5 Airy units as pinhole. Image processing was performed using LAS AF Lite Software (Leica Microsystems, Heidelberg, Germany). For each image, the fluorescence in the bleached region was normalized for the fluorescence of the background and the percentage of the initial fluorescence was calculated for each time point. FRAP recovery curves were represented (Appendix Fig S9) following the formula $\% \text{fluorescence} = \left( \frac{ROIt}{BGt} * \frac{ROIt_o}{BGt_o} \right) * 100$, where ROIt is the intensity in the selected ROI at time point t, BGt is the intensity in the background at time point $t$, ROIt$_0$ is the intensity in the selected ROI at time 0, and BGt$_0$ is the intensity in the background at time 0. Data were analyzed using GraphPad Prism Software (San Diego, USA).

### Nuclear fractionation

Nuclear fractionation from hippocampus of 30-week-old wild-type and R6/1 mice was performed using a Subcellular Protein Fractionation Kit for Tissues (Thermo Fisher Scientific, #87790) following manufacturer's instructions. Chromatin-bound and nuclear soluble fractions were obtained and examined by Western blot as described above.

### RNA-seq

RNA-seq data were generated from 30-week-old wild-type and R6/1 mice hippocampal tissue. RNA was isolated from 9 independent

biological replicates for each genotype using the RNeasy Plus Kit (Qiagen, #74136) according to the manufacturer's instructions. Quality assessment was performed using Bioanalyser eukaryotic total RNA nano series II chip (Agilent, #5067-1511), and all samples achieved a RNA integration number (RIN) between 9 and 10. Libraries were prepared from 9 biological replicates of each condition using the TruSeq Stranded mRNA Library Prep Kit (Illumina, #20020594) following manufacturer's instructions and sequenced using the HiSeq 2500 sequencing platform (Illumina, San Diego, USA).

### ChIP-seq

Lamin B1 ChIP-seq was performed as previously described (Sadaie et al, 2013) by using 30-week-old wild-type and R6/1 mice hippocampal tissue. Briefly, for each biological replicate, hippocampus from 5 mice was pooled together, homogenized in PBS supplemented with proteinase inhibitors (Sigma, #13317600), and posteriorly cross-linked with formaldehyde 1% for 15 min at room temperature. Cross-linking reaction was stopped by a 5 min of incubation with 2 M glycine, and the cross-linked material was washed 3 times with ice-cold PBS. Cells were lysed using cell lysis buffer (10 mM HEPES pH 8, 85 mM KCl, 0.5% NP-40), and nuclei were extracted using nucleus extraction buffer (0.5% SDS, 10 mM EDTA pH 8, 50 mM Tris–HCl pH 8). Purified nuclear fraction was subjected to sonication using Bioruptor Pico (Diagenode, Belgium) to obtain DNA fragments of 200–500 bp. Sonicated chromatin was incubated overnight at 4°C with anti-rabbit magnetic Dynabeads (Thermo Fisher Scientific, #11203D) pre-complexed with 10 μg of rabbit anti-lamin B1 antibody. After 6 washes with RIPA buffer (20 mM Tris–HCl (pH 7.5), 150 mM NaCl, 1 mM Na$_2$EDTA, 1 mM EGTA, 1% NP-40, 1% sodium deoxycholate, 2.5 mM sodium pyrophosphate, 1mM b-glycerophosphate, 1 mM Na3VO), chromatin was eluted and de-cross-linked by overnight incubation at 65°C, followed by a 30 min of RNase (Ambion, #AM2271) and 2 h of proteinase K (Thermo Fisher Scientific, #AM2548) treatments. DNA purification was carried out with MinElute PCR Purification Kit (Qiagen, #28006), and libraries were prepared using the NEBNext Ultra II DNA Library Prep Kit from Illumina (New England Biolabs, #37645) according to the manufacturer's instructions. DNA size selection was performed after PCR amplification using E-Gel Precast Agarose Electrophoresis System (Invitrogen, #A42100). Samples were sequenced single end using 50-bp reads on the HiSeq 2500 and HiSeq 4000 platforms (Illumina, San Diego, USA).

### ATAC-seq

ATAC-seq experiments were performed as previously described (Buenrostro et al, 2013), with slight modifications, in 3 independent biological replicates using hippocampal tissue from 25-week-old wild-type and R6/1 mice. Briefly, a frozen mouse hippocampus for each biological replicate was pulverized using a grinder and pestle settle on dry ice, and tissue powder was lysed in LB1 buffer (1M HEPES pH 7.5, 5 M NaCl, 0.5 M EDTA pH 8.0, 50% glycerol, 10% NP-40, 10% Triton X-100) for nuclear isolation. Approximately 50,000 nuclei were used for the transposition reaction using hyperactive Tn5 transposase (Illumina Cat, #FC-121-1030) followed by 13 cycles of PCR amplification. "Nucleosome free" and "mono-

nucleosome fragments" were obtained by size selection of DNA fragments between 170 bp and 400 bp using SPRIselect beads (Beckman Coulter, #B23319) before single-end sequencing to generate 50-bp reads on the HiSeq 4000 platform (Illumina, San Diego, USA).

### RNA-seq, ChIP-seq, and ATAC-seq data analysis

ChIP-seq samples were mapped against the mm9 mouse genome assembly using Bowtie with the option –m 1 to discard those reads that could not be uniquely mapped to just one region (Langmead *et al*, 2009). We ran the EDD tool (parameters: GAP = 5 and BIN SIZE = 37) to identify LADs on our ChIP-seq samples (Lund *et al*, 2014). Triplicates of each condition were pooled together, once a high degree of similarity in the set of reported LADs and target genes was confirmed among replicates. The UCSC genome browser was used to generate the screenshots of each group of experiments along the manuscript (Kent *et al*, 2002). The RNA-seq samples were mapped against the mm9 mouse genome assembly using TopHat (Trapnell *et al*, 2009) with the option –g 1 to discard those reads that could not be uniquely mapped in just one region. DESeq2 (Love *et al*, 2014) was run over nine replicates of each genotype to quantify the expression of every annotated transcript using the RefSeq catalog of exons and to identify each set of differentially expressed genes. ATAC-seq samples were mapped against the mm9 mouse genome assembly using Bowtie with the option –m 1 to discard those reads that could not be uniquely mapped to just one region, and with the option –X 2000 to define the maximum insert size for paired-end alignment (Langmead *et al*, 2009). Mitochondrial reads were removed from each resulting map, and down-sampling was applied to obtain the same number of mapped fragments per sample. Correlation between biological replicates in terms of peaks was assessed to ensure high reproducibility before pooling each set of triplicates. MACS was run with the default parameters but with the shift size adjusted to 100 bp to perform peak calling (Zhang *et al*, 2008). The genome distribution of each set of peaks was calculated by counting the number of peaks fitted on each class of region according to RefSeq annotations (O'Leary *et al*, 2016). Distal region is the region within 2.5 Kbp and 0.5 Kbp upstream of the transcription start site (TSS). Proximal region is the region within 0.5 Kbp upstream of the TSS. UTR, untranslated region; CDS, protein-coding sequence; intronic regions, introns; and the rest of the genome, intergenic. Peaks that overlapped with more than one genomic feature were proportionally counted the same number of times. For the generation of metaprofiles, seqMINER tool (Ye *et al*, 2011) was used in combination with ggplot2 package from R (https://ggplot2.tidyverse.org/) using an in-house script.

### Differential chromatin accessibility

Integrated analysis of ATAC-seq data was performed using the open Galaxy platform (https://usegalaxy.org/). A list of high confident peaks identified with MACS2 for each genotype was generated by selecting peaks found in at least 2 different replicates. For differential accessible region identification, edgeR galaxy tool was used with a merge of all high confident peaks identified in both genotypes applying the TMM method implemented in edgeR for normalization and dispersion calculation of the different biological samples. The results

were further filtered based on FDR < 0.05. Peaks were annotated to the closest TSS using Homer tool integrated in Galaxy platform.

### Motif analysis

For motif analysis, the Meme-ChIP suite (version 4.12.0) tool (Bailey *et al*, 2009) was used in differential enrichment mode together with Hocomoco (version 11 FULL) human and mouse PWMs. As input, 600-bp regions surrounding the summit of differential accessible peaks were used for motif discovery (using relaxed regions as control for compacted regions and vice-versa) and only motifs centrally enriched were considered.

### Gene ontology

For functional enrichments in biological processes (BP), DAVID (Huang *et al*, 2009) tool was used by providing closest genes ID obtained by Homer when using differentially accessible regions, with subsets of genes identified as differentially expressed or with genes located in LAD-identified regions. Terms with Benjamini's adjusted *P*-value < 0.05 were selected for bar graph representations.

### Data visualization

UCSC genome browser (Kent *et al*, 2002) was used for genome-wide visualization of ChIP-seq and ATAC-seq data.

### Statistics

Sample size was determined by using the power analysis method: 0.05 alpha value, 1 estimated sigma value, and 75% of power

**The paper explained**

**Problem**

Lamins, the major structural proteins within the nuclear lamina, are crucial for the functionality of the nucleus. Our previous results showed that lamin B levels are increased in a brain region-dependent manner in Huntington's disease (HD). However, it is not known whether this alteration has consequences for nuclear function of cells expressing mutant huntingtin and plays a role in HD pathophysiology.

**Results**

Lamins levels were analyzed in the cortex, striatum, and hippocampus of the R6/1 mouse model of HD and HD patients. We observed a brain region and age-dependent increase in lamin B1 levels. Through fluorescence-activated nuclear suspension imaging, we determined that lamin B1 levels are increased in specific neuronal populations, which is correlated with alterations in nuclear morphology and nucleocytoplasmic transport disruption as assessed by fluorescence recovery after photobleaching. In addition, ChIP-seq analysis in hippocampal nuclei from R6/1 mouse showed a partial unstructuring of lamin B1-associated domains and changes in chromatin accessibility (assessed by ATAC-seq), which correlates with transcriptional dysregulation determined by RNA-seq. In support of a significant role of lamin B1 in HD pathology, the administration of betulinic acid in R6/1 mouse partially restored lamin B1 levels and attenuated both motor and cognitive dysfunction.

**Impact**

Our work highlights increased lamin B1 levels as a new pathogenic mechanism for HD, providing a novel target for its intervention.

detection. Such analysis was chosen as the result of previous behavioral experiments in our laboratory. *N* values are given throughout the manuscript in the figure legends. Grubb's test was performed to determine the significant outlier values with pre-established criteria. All the results are expressed as the mean ± SEM. Statistical tests were performed using Student's *t*-test for one grouping variable and the one or two-way ANOVA for multi-component variables, followed by Bonferroni's or Turkey's post hoc test as indicated in the figure legends. Linear regression analyses were performed using R-squared. For non-parametric ChIP-seq and ATAC-seq data analysis, the Wilcoxon–Mann–Whitney test was used as indicated in figure legend. Additional statistical analysis for ChIP-seq, ATAC-seq, and RNA-seq data is indicated in their respective sequencing analysis section. A 95% confidence interval was used, and values with a $P < 0.05$ were considered as statistically significant.

## Data availability

Raw data and processed information of the ChIP-seq, ATAC-seq, and RNA-seq experiments generated in this article were deposited in the National Center for Biotechnology Information Gene Expression Omnibus (NCBI GEO) (Barrett *et al*, 2013) repository under the Accession Number GSE139884. Additionally, public datasets for hippocampal H3K9me3 (GSM2460430, https://www.ncbi.nlm.nih.gov/geo/query/acc.cgi?acc = GSM2460430) (Ding *et al*, 2017), H3K9ac (GSM2415914, https://www.ncbi.nlm.nih.gov/geo/query/acc.cgi?acc = GSM2415914) (Mews *et al*, 2017), and CTCF (GSM2228526, https://www.ncbi.nlm.nih.gov/geo/query/acc.cgi?acc = GSM2228526) (Sams *et al*, 2016) ChIP-seq experiments were retrieved for integrated data analysis.

**Expanded View** for this article is available online.

### Acknowledgements

We thank Ana López and Maria Teresa Muñoz for technical assistance; Isabel Crespo from Citomics Core Facility of the Institut d'Investigacions Biomèdiques August Pi i Sunyer (IDIBAPS) and Maria Calvo from the Advanced Microscopy Unit, Scientific and Technological Centers, University of Barcelona, for their support and advice concerning Cytometry and confocal techniques, respectively; Neurological Tissue Bank of the Biobank-Hospital Clínic-IDIBAPS for providing human brain tissue; Dr. Marcy MacDonald for the knock-in striatal cell line, Dr. Lucas for the N-mHtt-CFP plasmid; and Ana Saavedra for helpful discussions. Participant laboratories were supported by (1) EPN.: Ministerio de Economia y Competitividad, Spain (SAF2016-08573-R; PID2019-106447RB-100), and Fundación Ramón Areces (RL000607); (2) MN.: Cancer Research UK Cambridge Institute Core Grant (C9545/A29580); (3) LDiC: Ministerio de Economia, Indústria y Competitividad (BFU2016-75008-P and PID2019-108322GB-100); and (4) SP: Instituto de Salud Carlos III (ISCIII; PI18/00283), FERO Foundation, La Caixa Foundation (LCF/PR/PR12/51070001), and Cellex Foundation who provided research facilities and equipment; MGF was supported by a grant (FI-2016) from Agència de Gestió d'Ajuts Universitaris i de Recerca (AGAUR), and AP is a fellow of Sir Henry Wellcome (215912/Z/19/Z).

### Author contributions

RAV, MGF, and EPN conceptualized the study and interpreted the results, and EPN supervised it. RAV and MGF designed and performed most of the experiments with the help of CCP, JCM, AG, and KCV. MN conceptualized and contributed to the design of ChIP-seq, ATAC-seq, and RNA-seq experiments. YI and AP contributed with the design and performance of ChIP-seq, ATAC-seq, and RNA-seq experiments. GS and SS contributed in the analysis of ChIP-seq, ATAC-seq, and RNA-seq. EB analyzed and composed the figures for ChIP-seq, ATAC-seq, and RNA-seq experiments. LDiC and SP reviewed the content and contributed in the ChIP-seq, ATAC-seq, and RNA-seq experiments and conceptualization and interpretation of the results. RAV and MGF analyzed data and composed the figures. EPN secured the funding, the collaborations, and the execution of the entire project. RAV, MGF, and EPN wrote the manuscript. All the authors critically reviewed the content and approved the final version.

### Conflict of interest

The authors declare that they have no conflict of interest.

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
