## [Review Process File · EMBO Molecular Medicine]

Neuron type-specific increase in lamin B1 contributes to nuclear dysfunction in Huntington's disease

Rafael Alcalá-Vida, Marta Garcia-Forn, Carla Castany-Pladevall, Jordi Creus Muncunill, Yoko Ito, Enrique Blanco, Arantxa Golbano, Killian Crespí-Vázquez, Aled Parry, Guy Slater, Shamith Samarajiva, Sandra Peiró, Luciano Di Croce, Masashi Narita, and Esther Perez-Navarro
DOI: 10.15252/emmm.202012105

Corresponding author(s): Esther Perez-Navarro (estherperez@ub.edu)

Review Timeline:

Submission Date:	31st Jan 20
Editorial Decision:	25th Feb 20
Revision Received:	16th Oct 20
Editorial Decision:	8th Nov 20
Revision Received:	20th Nov 20
Accepted:	24th Nov 20

Editor: Jingyi Hou

Transaction Report:

Thank you for the submission of your manuscript to EMBO Molecular Medicine. We have now received feedback from the three referees who agreed to evaluate your manuscript. As you will see from the reports below, referee #1 and #2 are more supportive than referees #3, who is more reserved and raises a series of important issues. The main critical point is that referee #3 is not convinced that the data convincingly demonstrate that increased lamin B1 levels indeed contribute to HD pathogenesis. The second critical point is about the underlying mechanism that is not sufficiently developed (as commented by both referees #1 and #2). The referees do offer suggestions to improve and strengthen the conclusions and we would like to encourage you to address these comments as indicated.

In particular, during our cross-commenting process (in which the referees are given the chance to make additional comments, including on each other's reports), it became clear that:

In agreement with referee #3, referee #2 thinks that knockdown and overexpression of lamin B1 need to be performed, and a more detailed analyses of human samples as suggested by referee #3 is required. Referee #1 thinks that the following issues should be addressed: "clarifying the analysis of laminB1 in human and mouse neurons and glia, having a better analysis of nuclear structure (3D) as suggested by reviewer 3, and ensuring the various controls requested by the reviewers are performed at a rigorous level". Both referees think that an evaluation of the nuclear pore structure and LINC complex could be considered out of the scope of this study. Referee #3 added "While it may be beyond the scope of this current study to comprehensively evaluate the nuclear pore and LINC complex, the authors must acknowledge this possibility as previous publications have shown that cytoplasmic accumulations of nucleoporins are present in HD. Moreover, the authors do not provide any conclusive evidence that alterations in the nuclear lamins are responsible for alterations in LAD associations or any HD phenotypes evaluated given their lack of mechanistic experiments (and their drug being non specific as has been pointed out). Furthermore, this does not negate the fact that the authors provide little evidence to support their conclusions regarding lamin pathology as has been mentioned by all reviewers regarding the discrepancies between the text and images provided in the figures. Nuclear lamina and nuclear pore pathology has been well documented across multiple neurodegenerative diseases including HD. Therefore, for this manuscript to significantly contribute to and advance the knowledge of HD pathophysiology, a significant number of experiments and revisions are necessary to support the claims that alterations to lamin levels are relevant to the underlying biology of HD. First, the authors must provide conclusive evidence that lamin levels are altered and lamin pathology is present in HD. As pointed out in the initial review, the current data presented are far from convincing. Second, the authors must demonstrate a mechanistic link between lamin alterations and HD pathophysiology. As a first pass, these are easiest to accomplish in primary cultures. However, in vivo validation (ie genetic manipulation of lamin in HD mice) would lend additional mechanistic insights."

We will not ask you to perform in vivo validation experiment as suggested by referee #3, as this would likely not be feasible in a reasonable timeframe. However, all other concerns and comments of the referees need to be convincingly addressed to improve the conclusiveness.

***** Reviewer's comments *****

Referee #1 (Remarks for Author):

Here the authors demonstrate that lamin B1 levels are increased in neurons from a mouse model of HD and in immunoblots from HD patients. Importantly, they use ChIP-sequencing to demonstrate altered chromatin domains that correlate with altered transcription, which has been linked strongly to HD pathogenesis. Pharmacological treatment with betulinic acid partially reduces lamin B1 and rescues behavioral abnormalities in a transgenic mouse model of HD, linking the two. While implication of lamin pathology in neurodegeneration is not new in general (appropriately reviewed in

the Introduction), the findings are novel in the context of HD. The paper also makes advances by linking specific chromatin changes with lamin alterations. The major concern from the context of a broad readership is the lack of information as to how mutant Htt promotes increased lamin B1 levels. The authors also need to resolve the issue of whether lamin B1 is increased in patient neurons (see below).

Specific comments:

1. Introduction. Very clearly and appropriately written.
2. Figure 1c,d. It would be interesting to know if repeat expansion length correlated with lamin levels.
3. Figure 2,3. It does not appear that there is a consistent increase in lamin B1 levels in the patient neurons examined. This issue must be addressed before publication because it is inconsistent with the thesis of the manuscript and the interpretation of the western blot and immunostaining data in Figures 1 and 2. There is some attempt in the discussion to address this issue, but the issue is nonetheless unclear. The statement in the Discussion that the immunostaining reveals increased lamin B1 is not supported by quantitative data.
4. Figure 5,6,7. The authors have demonstrated alterations in LADs, chromatin and transcription. However, the authors could be clearer about the extent to which changes in LADs contribute to the other findings.
5. Figure 8. Betulinic acid is not specific. The authors should comment on their choice of drug and discuss the issue of specificity more clearly.

Referee #2 (Remarks for Author):

The study by Alcala-Vida and colleagues investigates the contribution of the disruption of the nuclear lamina to the pathophysiology of Huntington's disease (HD). In particular, they show that the levels of lamin B1 in specific neuronal types are higher in a mouse model of HD than in WT mice and are also increased in cells from HD patients. Multi-omic analysis, combining RNA-seq, ChIP-seq and ATAC-seq, unveiled some links between changes in lamin-associated chromatin domains and transcriptional alterations in hippocampal tissue. Finally, complementing the correlative evidence, the authors demonstrate that the treatment of HD mice with betulinic acid restored lamin B1 levels and prevented motor and cognitive dysfunction. All together, this represents an impressive and highly multidisciplinary piece of work. The experimental design of the study is robust and the data seem of good quality. The study also provides a novel perspective in the etiology of an important neurological disorder that remains incurable, pinpointing possible novel target for therapeutic intervention. However, some aspects of the study could be improved.

1. Several blots for lamin A and lamin C presented in Figure S1 seem saturated and therefore its quantification may not be reliable. The authors should present better images.
2. The images presented for R6/1 mice in Figure 2c do not appear to have the same quality than for controls. Is the central photo out of focus? The blue signal (DNA) seems weaker in R6/1 mice. Is this correct? Maybe the authors could provide better images or show the signal for the different channels instead of the merged one.
3. Complementing the FANSI method, the authors could quantify the differences in fluorescence intensity observed in the immunohistochemistry experiments presented in Figures 2 and S2.
4. The point above is particularly relevant for Figure 1d, because the conclusion derived from these

images is very relevant and should be better supported. The R6/1 mice represent a model for HD that particularly exacerbates mHtt-related pathology by the strong expression of a truncated mHtt transgene. However, the authors conclude that the changes in laminB1 do not correlate with mHtt aggregates. This is an unexpected finding that would need further investigation. Can the authors use FANSI to explore the correlation between mHtt aggregates and laminB1 alterations in a more systematic and reliable way?

5. The authors may consider including cerebellar neurons in their analysis as a negative control. Purkinje neurons, for example, have a large nucleus and may provide clear images to support the cell type-specific effect.

6. The authors claim in the title and some other paragraphs that the reported effect is neuron-specific but this conclusion is largely based on the negative result obtained in glial cells. A general statement about neuronal specificity should be based on more cell types. Also, the nucleus of glial cells is much smaller than the neuronal nuclei, which can make difficult the detection of changes in lamin B levels or invaginations. I would recommend the exploration of other cell types with a nucleus more similar in size and structure to those observed in pyramidal neurons.

7. The authors present a PCA for RNA-seq samples (Fig. S6d). They should do the same for ChIP-seq and ATAC-seq samples. The tracks presented in Figure S5d could suggest that R6/1 2 is much closer to WT than to the other R6/1 samples. Is this the case? Does this correlate with the severity of the phenotype in the groups of mice used to obtain these samples? (on the other hand, WT 2 seem closer to the R6/1 samples, but this would be easier to appreciate in the PCA graph or in a Pearson correlation heatmap).

8. Some graphs in Figures 5, 7 and S5 should be revised because the labels and numbers on the Y-axis are unreadable.

9. The authors indicate that they mapped their ChIP-seq, RNA-seq and ATAC-seq samples into mm9 (which was assembled in 2007). Is this information correct? And if so, why? Mm10 is available since December 2011. There is no reason to present the data using an outdated version of the genome. The authors should revise all the analyses using the current (for the last 8 years) version of the mouse genome.

10. The authors speculate about the mechanism underlying the proposed increase in stability (p.11), but they do not explore any of these mechanisms, which would represent a key development. For example, do the previously reported changes in PKC δ present the same cellular specificity than the changes in laminB1? Is the link preserved between R6/1 mice and humans? The authors could downregulate PKC δ (for example, using the same siRNA used in Rue et al., 2014) and explore the impact on laminB1 stability. This straightforward experiment would greatly strengthen their model.

Typos:

- p.5: ... 3b).As expected,

Referee #3 (Comments on Novelty/Model System for Author):

In this manuscript, Alcalá-Vida et al attempt to define the pathophysiology of lamin alterations in

HD. While the authors use multiple experimental approaches to address how changes in the nuclear lamina may contribute to HD pathogenesis, much of the data presented are not convincing at all. The conclusions and interpretations are not conclusively supported by the data as it is currently presented in this manuscript. There are a number of very significant concerns detailed in the review that must be addressed before publication could be considered. Therefore, the reviewer does not support the publication of this manuscript.

Referee #3 (Remarks for Author):

In this manuscript, Alcalá-Vida et al attempt to define the pathophysiology of lamin alterations in HD. While the authors use multiple experimental approaches to address how changes in the nuclear lamina may contribute to HD pathogenesis, much of the data presented are not convincing. The conclusions and interpretations are not conclusively supported by the data as it is currently presented in this manuscript.

1. For experiments evaluating the VS stages of HD, the authors need to present each stage individually instead of combining I/II and III/IV. In many of the figures, the pathology seen in these combined stage groups does not apply to both (e.g. III pathology is different than IV). Therefore, the data as presented is difficult to interpret.
2. For human tissue pathology, have the authors conclusively ruled out that the alterations they are seeing in the nuclear lamina are not simply reflective of aging? It is known that lamin alterations and invaginations accumulate with age. Therefore, are the pathologies displayed by the authors truly pathogenic or part of the normal aging process? The authors need to have multiple controls with full z plane analytics to overcome the prevailing science that shows nuclear invaginations typify aging.
3. The mouse histology images presented are not convincing. It is very clear that the images presented are often from different depths within the mouse brain based on the morphology of the DG. This makes interpretation extremely complicated. The zoomed panels of neuronal nuclei make it increasingly clear that the authors are imaging vastly different z depths in their HD animals compared to controls. In many images, the control seems to be taken from a central section of a nucleus (where one can see a nice lamin ring). However, in the HD mice, the images appear to be from the top or bottom face of nuclei as one can observe the normal folds of the neuronal nuclear lamina. This is supported by vastly different NeuN intensities. Moreover, zoomed images of the CA1 region often show a different lamin intensity than that in the overview image- can the authors please reconcile this?
4. From both the histology and the fluorescence activated nuclear suspension imaging, it is unclear whether these "folds" and invaginations are truly pathological. The authors appear to present a single Z slice which is very misleading-and inappropriate when examining the 3d structure of the nucleus. Moreover, there are normally many folds in the neuronal nuclear lamina. Therefore, it is unclear if the authors were to examine more nuclei if these phenotypes would truly represent pathology. The authors must examine the full volume of the nucleus and not simply cutting thru the middle - as that will miss surface folding
5. The dextran assay as described is not an appropriate assessment of passive NCT and NPC permeability. The protocol described uses Triton for nuclear isolation. Triton alone can effectively permeabilize nuclear pores, thus allowing dextran to leak in. Nuclei should be isolated without Triton. Further, the authors must evaluate additional size dextrans (10kD, 70kD, 500kD) to support

any conclusions regarding the permeability of the NPC. Moreover, given the appropriate experimental set up, FRAP is not needed to evaluate the passive permeability properties of the nucleus.

6. Overall, the authors fail to provide any conclusive evidence that increased lamin levels contribute to the pathogenesis of HD. Their drug studies result in incredibly subtle changes in lamin levels. Moreover, the main readout from these studies is mouse behavior. However, behavioral rescues could be independent of the subtle effects on lamin levels. These studies are in no way tied back to the molecular defects presented throughout the course of this manuscript. The authors should evaluate the effects of lamin OE in primary neuronal cultures to determine if molecular alterations are reminiscent of those seen in HD. Moreover, does KD of lamin in primary HD neuronal cultures mitigate relevant phenotypes?

7. The nuclear lamina is tightly linked to nuclear pores. In fact many nucleoporins actively coordinate genome organization and transcription. However, the authors assume all transcriptional alterations are a result of subtle and variable increases in lamin expression. This reflects a lack of scientific understanding of nuclear function and the many proteins and structures in the nuclear membrane- eg. The nuclear pore (and its 30 different proteins) or the LINC complex. It is quite cavalier, and very likely wrong, to assume that observed HD phenotypes (in particular those relating to membrane permeability and transcription) are solely the effect of increases in lamin. To be comprehensive, the authors would have to examine the changes in the nuclear pore structure (e.g the many nucleoporins) and the linc complex as well- as these may be altered in HD (as others have already published)

Referee #1 (Remarks for Author):

Here the authors demonstrate that lamin B1 levels are increased in neurons from a mouse model of HD and in immunoblots from HD patients. Importantly, they use ChIP-sequencing to demonstrate altered chromatin domains that correlate with altered transcription, which has been linked strongly to HD pathogenesis. Pharmacological treatment with betulinic acid partially reduces lamin B1 and rescues behavioral abnormalities in a transgenic mouse model of HD, linking the two. While implication of lamin pathology in neurodegeneration is not new in general (appropriately reviewed in the Introduction), the findings are novel in the context of HD. The paper also makes advances by linking specific chromatin changes with lamin alterations. The major concern from the context of a broad readership is the lack of information as to how mutant Htt promotes increased lamin B1 levels. The authors also need to resolve the issue of whether lamin B1 is increased in patient neurons (see below). Specific comments:

1. Introduction. Very clearly and appropriately written.

Thank you very much for your kind comments.

2. Figure 1c,d. It would be interesting to know if repeat expansion length correlated with lamin levels.

Following the reviewer's suggestion, we performed a linear regression analysis to evaluate the correlation between the number of CAG repeats and lamin B1 protein levels. This analysis was only performed in samples from HD patients as we did not have the information about the number of CAG repeats from non-affected individuals' samples. Results obtained showed that there is not significant correlation between the number of CAG repeats and lamin B1 protein levels in the striatum, hippocampus and cortex from HD patients. These results have been included in the revised manuscript as Appendix Figure S2 and results section has been re-written accordingly (page 4, lines 3-8; see text highlighted in yellow).

3. Figure 2,3. It does not appear that there is a consistent increase in lamin B1 levels in the patient neurons examined. This issue must be addressed before publication because it is inconsistent with the thesis of the manuscript and the interpretation of the western blot and immunostaining data in Figures 1 and 2. There is some attempt in the discussion to address this issue, but the issue is nonetheless unclear. The statement in the Discussion that the immunostaining reveals increased lamin B1 is not supported by quantitative data.

We understand the reviewer's concern and we apologize if we did not address this issue clearly in the discussion. We stated that quantification of lamin B1 intensity in the nuclei from the putamen of HD patients by FANSI (Figure 3b initial submission) was not representative, as we only detected 10 neuronal nuclei in average for each sample. We considered that this number of nuclei analyzed is not sufficient to generate a robust conclusion. We speculate that nuclear abnormalities observed potentiate MSNs loss after processing of the tissue, probably due to repeated centrifugation, besides the neurodegenerative process itself. This could result in the evaluation of those MSNs with a healthier nuclear envelope (including "normal" lamin B1 levels) and not all the MSNs

remaining in the tissue. In order to overcome this possible problem and in order to clarify these results and in response to the reviewer's comment, we have now quantified lamin B1 intensity in the immunostainings. In agreement with the results obtained by western blot (Figure 1c initial submission, Figure EV2 in the revised manuscript), we detected an increase in lamin B1 levels in the nuclei of MSNs from HD patient's putamen at VS III-IV, but not in those at VS I-II. By using this technique, we were able to analyze lamin B1 levels in 50 nuclei/sample in average, in contrast to the 10 nuclei/sample that were analyzed by using FANSI, indicating that these results are more robust than those obtained by FANSI. In addition, by analyzing all the z stacks as suggested by referee #3, we also detected altered nuclear morphology in striatal neurons from VS III-IV HD patients. We have included these quantifications as Figure 4D in the revised manuscript and moved Figure 3b (initial submission) to supplementary material (Appendix Figure S8A) to avoid further confusion regarding this issue. Material and methods (page 19, lines 18-20 and 25-30), results (page 5, lines 32-33; page 6, lines 1-4) and discussion (page 12, lines 27-33; page 13 lines 1-3) sections have been modified accordingly (text highlighted in yellow).

4. Figure 5,6,7. The authors have demonstrated alterations in LADs, chromatin and transcription. However, the authors could be clearer about the extent to which changes in LADs contribute to the other findings.

Recent works have established that changes in the configuration of LADs are accompanied by changes in the chromatin accessibility and gene expression (Pascual-Reguant et al., 2018, Nat Commun. 24, 9:3420). Similarly, although our work reports subtle differences between LAD configuration in R6/1 vs wild-type mouse hippocampal nuclei at 30 weeks of age, these alterations are relevant and correlate with changes in the expression of genes related to nucleosome assembly. In addition, we have identified changes in the chromatin conformation and in the transcriptome, which include several transcription factors that play an important role in the central nervous system. Importantly, we have included into our study multiple bioinformatics analysis that involve data from more than one type of experiments. For instance, up- and down-regulated R61/WT genes in terms of expression show a similar trend in terms of chromatin accessibility, while genes associated to differentially ATAC-seq accessible peaks present a clear correlation between open/close chromatin levels and gene expression. Even though we have not observed dramatic changes in terms of gene expression and accessibility in the genes involved in reorganization of LADs between WT and R6/1 mouse hippocampus, we do have reported that genes located in differentially accessible ATAC-seq peaks present a correlation between laminB1 ChIP-seq levels and chromatin compaction. We consider that this constitutes an interesting link between alterations in LADs and chromatin accessibility and gene expression. Therefore, in sum, our results suggest that changes in the configuration of LADs, although subtle, could contribute to alter chromatin accessibility and gene expression in R6/1 mice hippocampal cells. In the revised version of the manuscript, we have rewritten the abstract (line 9) and the discussion (page 14, lines 13-18, 21-34; page 15, lines 1-3) sections to clarify this issue (text highlighted in yellow).

5. Figure 8. Betulinic acid is not specific. The authors should comment on their choice of drug and discuss the issue of specificity more clearly.

We acknowledge referee's concern about the choice of drug. Our aim was to use a drug that could directly target lamin B1 and that was therapeutically feasible. Even though directly targeting lamin B1 expression by using a shRNA or siRNA against *Lmnb1* would be the most specific way of downregulating lamin B1, this treatment would be highly invasive, and feasible in HD mouse models but not in patients in the near future. Among the few existent drugs with an effect on lamin B1 (Chiarini et al., 2008, *Biochim Biophys Acta* 1783:1642; Adam et al., 2013, *Nucleus*, 4:142; Hendrayani et al., 2013, 15:631; Li et al., 2013, *Clin Cancer Res* 19:4651; Hu et al., 2016, *Sci Rep* 6:31298) betulinic acid seemed the most appropriate drug to use. It was previously reported to directly target *Lmnb1* expression (Li et al., 2013, *Clin Cancer Res* 19:4651), to cross the blood brain barrier (Udeani et al., 1999, *Biopharm Drug Dispos* 20:379) and to have positive effects in the brain (Kaundal et al., 2018, *Eur J Pharmacol* 832:56; Navabi et al., 2018, *Behav Brain Res* 30:337; Lu et al., 2011, *Nitric oxide* 24:132; Li et al., 2018, *Front Mol Neurosci* 11:98). We agree with the reviewer that in addition to regulate lamin B1 levels, betulinic acid has been shown to have many other effects (Rios and Máñez, 2018, *Planta Med*, 84:8) and therefore, we cannot discard that some of the beneficial effects we observed could be due to the modulation of other altered mechanisms in cells expressing mHtt. Yet, we are confident that the improvement in nuclear functionality observed in hippocampal neurons is the result of the effects of betulinic acid on lamin B1 levels, since no beneficial effects were detected in striatal neurons in which we did not observe lamin B1 levels normalization. Thus, we would like to note that while betulinic acid is not specific for lamin B1, we feel positive for the translational potential of this drug. Biopharmaceutical innovation is increasingly focused on the potential of multispecific drugs for treating disorders with multiple dysregulated mechanisms (Deshaies 2020, *Nature* 580:329). Although HD is a monogenic disorder, drugs that specifically target the cause of the disease do not exist. Until this kind of drug is developed, multispecific drugs targeting multiple altered mechanisms are the best option for its treatment. Thus, we feel confident about the potential of betulinic acid to fulfill this aim. In the revised version of the manuscript, we have discussed this issue (page 15, lines 24-28; text highlighted in yellow).

Referee #2 (Remarks for Author):

The study by Alcalá-Vida and colleagues investigates the contribution of the disruption of the nuclear lamina to the pathophysiology of Huntington's disease (HD). In particular, they show that the levels of lamin B1 in specific neuronal types are higher in a mouse model of HD than in WT mice and are also increased in cells from HD patients. Multi-omic analysis, combining RNA-seq, ChIP-seq and ATAC-seq, unveiled some links between changes in lamin-associated chromatin domains and transcriptional alterations in hippocampal tissue. Finally, complementing the correlative evidence, the authors demonstrate that the treatment of HD mice with betulinic acid restored lamin B1 levels and prevented motor and cognitive dysfunction. All together, this represents an impressive and highly multidisciplinary piece of work. The experimental design of the study is robust and the data seem of good quality. The study also provides a novel perspective in the etiology of an important neurological disorder that remains incurable,

pinpointing possible novel target for therapeutic intervention. However, some aspects of the study could be improved.

1. Several blots for lamin A and lamin C presented in Figure S1 seem saturated and therefore its quantification may not be reliable. The authors should present better images.

We thank the reviewer for this observation. According to his/her suggestion representative blots in Figure S1 (Now Fig. EV1) have been replaced by better images.

2. The images presented for R6/1 mice in Figure 2c do not appear to have the same quality than for controls. Is the central photo out of focus? The blue signal (DNA) seems weaker in R6/1 mice. Is this correct? Maybe the authors could provide better images or show the signal for the different channels instead of the merged one.

We thank the reviewer for this observation. The quality of images presented in Figure 2c (initial submission) was probably decreased during the generation of the final version submitted. Now, we have corrected them accordingly, and as suggested, we have included the different channels separately to facilitate their visualization (Figure 3A in the revised manuscript).

3. Complementing the FANSI method, the authors could quantify the differences in fluorescence intensity observed in the immunohistochemistry experiments presented in Figures 2 and S2.

We decided to set up the FANSI method to analyze lamin B1 levels and nuclear morphology since it was very difficult to individually quantify lamin B1 intensity and nuclear structure in nuclei from mouse brain, as they are highly packed. Furthermore, FANSI method is more accurate as it allows quantifying fluorescence intensity in the entire nuclei and not only in a plane like by conventional microscopy. Although in Figure S2 (initial submission, now Appendix Figure S6) nuclei are isolated and would be more plausible to analyze them than in the tissue, the aim of this figure was to show the isolation of the different cell types immunostained with their respective markers, prior to the analysis by FANSI. FANSI allows to analyze 5,000 nuclei/sample in average, which we consider that generates a more robust conclusion as the number of nuclei analyzed by using this technique is much higher than by using conventional microscopy. Thus, we only used these immunostainings to support qualitatively western blot data. In addition, these immunostainings allowed us to identify nuclear morphology alterations. Regarding analysis of lamin B1 levels in nuclei from human putamen, and in response to referee #1 and referee #3, we decided to quantify fluorescence intensity and nuclear morphology in tissue as we only detected 10 neuronal MSN nuclei in average for each sample when analyzed by FANSI and we considered that this number of nuclei was not enough to generate a robust conclusion. We speculate that nuclear abnormalities observed potentiate MSNs loss after processing of the tissue, probably due to repeated centrifugation, besides the neurodegenerative process itself. This could result in the evaluation of those MSNs with a healthier nuclear envelope (including “normal” lamin B1 levels) and not all the MSNs remaining in the tissue. In agreement with the results

obtained by western blot (Figure 1 initial submission, now Figure EV2), we detected an increase in lamin B1 levels in the nuclei of MSNs from HD patient's putamen at VS III-IV, but not in those at VS I-II, which was accompanied by altered morphology. By using this technique, we were able to analyze lamin B1 levels in 50 nuclei/sample in average, in contrast to the 10 nuclei/sample that were analyzed by using FANSI. We have included this quantification as Figure 4D in the revised manuscript and moved Figure 3b (initial submission) to supplementary material as Appendix Figure S8A. Material and methods (page 19, lines 18-20 and 25-30), results (page 5, lines 32-33; page 6, lines 1-4) and discussion (page 12, lines 27-33; page 13, lines 1-3) sections have been modified accordingly (text highlighted in yellow).

4. The point above is particularly relevant for Figure 1d, because the conclusion derived from these images is very relevant and should be better supported. The R6/1 mice represent a model for HD that particularly exacerbates mHtt-related pathology by the strong expression of a truncated mHtt transgene. However, the authors conclude that the changes in laminB1 do not correlate with mHtt aggregates. This is an unexpected finding that would need further investigation. Can the authors use FANSI to explore the correlation between mHtt aggregates and lamin B1 alterations in a more systematic and reliable way?

Following the reviewer's suggestion, we used FANSI to explore the correlation between mHtt aggregates and lamin B1 alterations in the striatum of R6/1 mice at 30 weeks of age. Similarly to what we observed by immunohistochemistry, results obtained by using FANSI show that alterations in lamin B1 intensity and nuclear circularity take place in nuclei with and without mHtt inclusions. In addition, our results showing that treatment with betulinic acid restores lamin B1 levels and improves nuclear morphology and transport in CA1 hippocampal neurons nuclei, but does not have any effect on the density and area of mHtt aggregates, reinforce the idea that these two events are taking place independently. This data has been included in the revised manuscript as Figure 4C to further support the conclusion derived from Figure 1d (initial submission) that has been moved to supplementary material as Appendix Figure S5 in the revised manuscript. Results (page 5, lines 9-12) and discussion (page 13, lines 9-12; page 13, lines 31-33; page 15, lines 10-13) sections have been modified accordingly (text highlighted in yellow).

5. The authors may consider including cerebellar neurons in their analysis as a negative control. Purkinje neurons, for example, have a large nucleus and may provide clear images to support the cell type-specific effect.

Following the reviewer's suggestion, we quantified by Western blot lamin B1 protein levels in the cerebellum of 30-week-old wild-type and R6/1 mice and found an increase in R6/1 mice in comparison to wild-type mice cerebellum (Referee #2 Figure 1A). In order to determine whether this increase was present in Purkinje neurons, we performed a double immunohistochemistry with anti-lamin B1 and anti-calbindin antibodies. Here, we show representative z-projections from 30 week-old wild-type and R6/1 mice (Referee #2 Figure 1B). As it can be appreciated in the images, the high compactness of cerebellar cells did not allow us to specifically quantify nuclear morphology in the Purkinje population although it seemed altered (Referee #2 Figure 1B). However, quantification of lamin B1 intensity was possible and results obtained

showed an increase in lamin B1 levels in Purkinje neurons nuclei from R6/1 in comparison to wild-type mouse (Referee #2 Figure 1B). Previous studies have already shown that cerebellum, and specifically Purkinje neurons, is also affected in HD patients (Rodda et al., 1981, *J Neurol Sci* 50:147; Jeste et al., 1984, *Exp Neurol* 85:78; Rüb et al., 2013, *Brain Pathol* 23:165; Malvinder et al., 2019, *Ann Neurol* 85:396) and mouse models (Tourmaine et al., 2000, *Proc Natl Acad Sci USA* ; 97:8093; Dougherty et a., 2012, *Exp Neurol* 236:171; 2013, *Exp Neurol* 240:96) and thus, cerebellum would not serve as a good negative control. Therefore, we decided to do not introduce this data in the revised version of the manuscript, as it can be confusing.

Figure 1. Figures for Referee not shown.

6. The authors claim in the title and some other paragraphs that the reported effect is neuron-specific but this conclusion is largely based on the negative result obtained in glial cells. A general statement about neuronal specificity should be based on more cell types. Also, the nucleus of glial cells is much smaller than the neuronal nuclei, which can make difficult the detection of changes in lamin B levels or invaginations. I would recommend the exploration of other cell types with a nucleus more similar in size and structure to those observed in pyramidal neurons.

We apologize for the misunderstanding generated with the repeated “neuron-specific” use, including in the title. We understand that this statement is confusing since it seems to be referring to all types of neurons. What we wanted to point out is that changes in lamin B1 and nuclear function only occur in certain types of neurons.

In the hippocampus, we analyzed two comparable neuronal populations in terms of nuclear size and found that CA1, but not DG neurons, presented lamin B1 alterations. To assess further the cell-type specificity in neurons with a nucleus more similar in size to that of striatal MSN, we used the FANSI to analyze lamin B1 intensity and nuclear circularity in striatal NeuN+/Ctip2- neurons. This population corresponds to striatal interneurons, which have a nuclear diameter between 6-11 μm that is comparable to the size of MSN’s nucleus (10-11 μm). Our results show that NeuN+/Ctip2- striatal neurons from R6/1 mouse at 30 weeks of age do not present lamin B1 alterations (shown as Figure 3B in the revised manuscript). Results section has been modified accordingly (page 5, lines 23-24 and 26; text highlighted in yellow). Here (Referee #2 Figure 2), we show representative images from 30 week-old wild-type and R6/1 mice just to compare the nuclear size of striatal MSNs and interneurons.

Referee #2 Figure 2. Representative images showing NeuN+/Ctip2+ and NeuN+/Ctip2- nuclei in the striatum of 30-week-old wild-type (WT) and R6/1 mice. Scale bar 7 μm .

In addition, and in order to be sure that glial cells expressing mHtt do not present alterations in lamin B1 levels, we analyzed lamin B1 by immunohistochemistry in striatal oligodendrocytes. In line with the results obtained by FANSI, 30-week-old R6/1 mouse oligodendrocytes did not show an increase in lamin B1 neither alterations in nuclear morphology (Appendix Figure S3 in the revised manuscript) in comparison with wild-type mouse. In the revised manuscript, results section has been modified accordingly (page 5, lines 1-2; text highlighted in yellow).

Altogether, we consider that the results obtained so far when analyzing R6/1 mouse brain show that alterations in lamin B1 are only present in some neuronal populations, but not in glial cells. Among neurons, only some types are affected. Thus, we conclude that lamin B1 alterations in HD are occurring in a neuronal-type specific manner. Thus, we have replaced “neuron-specific” by “neuron-type specific” along the manuscript and in the title.

7. The authors present a PCA for RNA-seq samples (Fig. S6d). They should do the same for ChIPseq and ATAC-seq samples. The tracks presented in Figure S5d could suggest that R6/1 2 is much closer to WT than to the other R6/1 samples. Is this the case? Does this correlate with the severity of the phenotype in the groups of mice used to obtain these samples? (on the other hand, WT 2 seem closer to the R6/1 samples, but this would be easier to appreciate in the PCA graph or in a Pearson correlation heatmap).

The EDD tool identifies LADs as genomic domains with a high density of laminB1 peaks furnished with a limited number of weak signals within. Depending on the zoom, visualization of the LADs on the genome browsers produces variable results as the resolution of the screen to show the peaks and gaps constituting one LAD is limited. Thus, we agree that it is difficult to conclude about relationships between samples from a particular screenshot taken with a particular degree of zoom. In addition, phenotyping information from animals comprising genomic experiments was not evaluated, as all experiments were conducted with tissue collected at basal state. In order to clarify this issue, we have performed an entirely new bioinformatics analysis at distinct levels of information: LaminB1/Input ChIP-seq signal, LAD block overlaps and matching of genes annotated inside LADs.

First, we have used the UCSC table browser on the tracks shown in Figure S5d (initial submission, Appendix Figure S10E in the revised manuscript) to calculate all of the pairwise correlation coefficients of the triplicates of WT and R6/1 (Referee #2 Figure 3, left). Apparently, WT samples are comparable while R6/1 samples seem to be more heterogeneous. Caution must be exerted, though, on this type of genome-wide results. In fact, when running the same tool on the RNA-seq experiments, we have obtained similar correlation values between samples of distinct conditions (Referee #2 Figure 3, right).

Referee #2 Figure 3. Pairwise correlation coefficients calculated with the UCSC table browser over ChIP-seq of LaminB1 against the corresponding inputs and RNA-seq profiles of WT and R6/1 (parameters: chromosome 19, bin size=10 kbp, CC (r) shown as a value between 0 and 100).

Moreover, we have performed the PCA analysis of WT and R6/1 samples using the values of the LaminB1/Input ratio at bin level (Referee #2 Figure 4). Probably due to the nature of this signal (LADs are series of non-contiguous LaminB1 peaks), the resulting plot is not conclusive taking into account the correlation heat maps above. Importantly, in this analysis, we have not observed that R6/1 2 is closer to WT samples or WT 2 is near of the R6/1 group. On the contrary, both samples are classified into different areas of the plot.

Referee #2 Figure 4. PCA analysis of the triplicates of LaminB1/Input in WT and R6/1 conditions (parameters: chromosome 19, bin size =100 bp).

To overcome the limitations of the analysis at signal level, we have performed an alternative analysis based on the overlap between LADs of each replicate (Referee #2 Figure 5 and Appendix Figure S10D). Venn diagrams of the overlaps indicate again that WT samples are comparable, while R6/1 samples are more heterogeneous.

Referee #2 Figure 5. LADs identified by the EDD software at each ChIP-seq replicate.

The heat map of overlap percentages indicates that samples of each group are classified with their counterparts (Referee #2 Figure 6). We have noticed, though, that the R6/1 R2 sample could be clustered together with WT experiments. These results can be biased for the overlap definition (at least one nucleotide is shared between two LADs), which can produce that two LADs with minimum overlap are considered to entirely match. Therefore, it is necessary to work with another methodology beyond ChIP signal quantification or LAD overlap that ensures higher resolution when classifying two LADs (i.e. matching genes in common inside two LADs, see below) to confirm or not previous results.

Referee #2 Figure 6. Heat map of percentage of overlap (left) and PCA (right) of LADs in the triplicates.

Therefore, to improve the accuracy of the comparisons in our assessment, we have decided to perform comparisons between LADs at gene level. First, we have identified the genes annotated inside the LADs reported at each replicate of WT and R6/1. Next, we have performed PCA analysis of the ChIP-seq replicates by using the genes in the genome that belong to the LADs found in these ChIP-seq experiments. Remarkably, both WT and R6/1 samples are clearly separated into two different groups in the resulting PCA plot (Referee #2 Figure 7).

Referee #2 Figure 7. PCA analysis of ChIP-seq replicates using the inclusion of genes in one or more of the LADs as identified at each ChIP-seq WT and R6/1 experiment.

Taking into account this final analysis, we consider that our set of ChIP-seq LaminB1 experiments is consistent between each class of replicates and we have included this PCA plot (Referee #2 Figure 7) as Appendix Figure S10C in the revised manuscript.

For our ATAC-seq experiments, which contain sharp peaks different from LAD-like blocks, we have performed PCA analysis of the read counts determined on the promoters of the full set of genes in the genome. The resulting PCA plot shows that the replicates of each class are clustered separately (Referee #2 Figure 8, left). Interestingly, when performing the PCA only over the genes previously reported as Up-regulated and Down-regulated in the transcriptomic analysis of WT Vs. R6/1, both groups of ATAC-seq samples are still more neatly distinguished (Referee #2 Figure 8, right). This finding indicates there is a clear link between our experiments on chromatin accessibility and our expression data.

Referee #2 Figure 8. PCA analysis of ATAC-seq samples with the full set of genes in the genome (left) and using the Up/Down-regulated genes of the RNA-seq experiments (right).

We have added the PCA plot of ATAC-seq samples in Appendix Figure S11B of the revised manuscript.

8. Some graphs in Figures 5, 7 and S5 should be revised because the labels and numbers on the Y-axis are unreadable.

We thank the reviewer for this observation. Following his/her recommendation, we have corrected Y-axis of Figures 5, 7 and S5 to improve their visualization.

9. The authors indicate that they mapped their ChIP-seq, RNA-seq and ATAC-seq samples into mm9 (which was assembled in 2007). Is this information correct? And if so, why? Mm10 is available since December 2011. There is no reason to present the data using an outdated version of the genome. The authors should revise all the analyses using the current (for the last 8 years) version of the mouse genome.

It is common to find in the literature that for many species a previous genome release is still more frequently used than other versions released later. In fact, when consolidated international consortiums of research (e.g. ENCODE or Roadmap Epigenomics) decide to deliver large-scale genomic data in a particular release, that version of the genome becomes the default assembly for a long time, irrespectively of the appearance of a new version. This has to do also with the generation of a richer collection of genomic and epigenomic annotations associated to the genome release that was selected as a reference for those consortiums by the rest of the scientific community. Consequently, for many genomes there is a period of time in which more than one assembly is accepted as a standard version (e.g. mm9 and mm10 for mouse,

hg19 and hg38 for human or dm6 and dm3 for the fruit fly, see Referee #2 Figure 9). In our case of study, the collection of tracks provided by popular genome browsers such as UCSC is still richer for mm9 than for mm10. For instance, we integrated our data in several figures of the manuscript with the UCSC *laminB1* super track, which contains the Van Steensel lab LaminB1 DamID LADs (Peric-Hupkes et al., 2010, Mol Cell 38:603). This track is, though, not available in the mm10 browser. For all these reasons, we consider that all our bioinformatics analyses performed on mm9 are consistent, valid, and agree well with the current state of the art in our field in which this genome release is frequently used on genome-wide analysis.

Referee #2 Figure 9. The UCSC browser provides direct links to hg19/hg38 and mm9/mm10 in its homepage (screenshot taken on Sunday 3/29/2020).

10. The authors speculate about the mechanism underlying the proposed increase in stability (p.11), but they do not explore any of these mechanisms, which would represent a key development. For example, do the previously reported changes in PKC δ present the same cellular specificity than the changes in laminB1? Is the link preserved between R6/1 mice and humans? The authors could downregulate PKC δ (for example, using the same siRNA used in Rue et al., 2014) and explore the impact on laminB1 stability. This straightforward experiment would greatly strengthen their model.

As the reviewer points out, knowing the mechanism/s leading to increased lamin B1 levels in cells expressing mHtt would represent a key development, as it could be also a target for therapeutic intervention. As also noticed by the reviewer, we have previously shown that PKC δ levels are decreased in a brain region-dependent manner in R6/1 mouse (Rue et al., 2014, Neuromol Med 16:25). However, we did not analyze cellular specificity as the labelling of PKC δ by immunohistochemistry, at least with the antibodies we have used, shows a very faint and diffuse labeling that difficulties co-localization analysis. Following the reviewer's suggestion, we knocked down PKC δ in striatal cells overexpressing mHtt and analyzed lamin B1 levels. Our results show that, in these experimental conditions, lamin B1 levels were higher than in cells overexpressing mHtt with normal levels of PKC δ . Moreover, we examined PKC δ levels in the same samples where lamin B1 levels were analyzed and made a correlation between these two parameters. We observed that in the striatum, but not in the hippocampus, there is a nice correlation between PKC δ and lamin B1 levels. These results were not unexpected since when we used betulinic acid to treat mice we observed a decrease in lamin B1 levels in the hippocampus, but not in the striatum, of R6/1 mouse suggesting that different mechanisms are leading to increased lamin B1 levels in these brain regions. Therefore, we suggest that downregulation of PKC δ levels in the striatum of R6/1 mice could be playing a role in increasing lamin B1 levels. These results have been included in the revised version of the manuscript as Figure EV3. Material and methods (page 17, lines 20-28; page 18, lines 9-11; page 19, lines 1-10, 18-20 and 30-32), results (page 5,

lines 9-23) and discussion (page 12, lines 14-19) sections have been modified accordingly (text highlighted in yellow).

Typos: - p.5: ... 3b).As expected,

We thank the reviewer for these observations. Typos have been revised and modified accordingly.

Referee #3 (Remarks for Author):

In this manuscript, Alcalá-Vida et al attempt to define the pathophysiology of lamin alterations in HD. While the authors use multiple experimental approaches to address how changes in the nuclear lamina may contribute to HD pathogenesis, much of the data presented are not convincing. The conclusions and interpretations are not conclusively supported by the data as it is currently presented in this manuscript.

1. For experiments evaluating the VS stages of HD, the authors need to present each stage individually instead of combining I/II and III/IV. In many of the figures, the pathology seen in these combined stage groups does not apply to both (e.g. III pathology is different than IV). Therefore, the data as presented is difficult to interpret.

We agree with the reviewer that the presentation of each stage individually would be more informative than combining them. However, the number of available HD brains' samples in the Neurological Tissue Bank of the Biobank-Hospital Clínic-IDIBAPS was limited. We had only one sample from VS-I, thus combining it with VS-II seemed the best option, since it would be difficult to make any conclusion if plotted individually. For each of the other VS, we had less than 3 samples (see Appendix Table S1). Since the availability of samples was limited, we decided to broadly classify them as I/II (early pathology) and III/IV (advanced pathology) when representing our data. In order to rule out any confounding effect due to the combining of the stages, we show Referee #3 Figure 1. This figure shows that lamin B1 protein levels do not differ significantly in the putamen of HD patients at VS-III – IV. Thus, we consider that the combining of these stages is not masking the interpretation of the results.

Referee #3 Figure 1. Lamin B1 levels in the putamen of HD patients at VS-III – IV do not differ significantly. Violin plot showing the distribution of lamin B1 protein levels of individual putamen samples from HD patients at Vonsattel stages (VS) III (circles), III/IV (square) and IV (triangles).

2. For human tissue pathology, have the authors conclusively ruled out that the alterations they are seeing in the nuclear lamina are not simply reflective of aging? It is known that lamin alterations and invaginations accumulate with age. Therefore, are the pathologies displayed by the authors truly pathogenic or part of the normal aging process? The authors need to have multiple controls with full z plane analytics to overcome the prevailing science that shows nuclear invaginations typify aging.

We agree with the referee that lamin alterations and invaginations accumulate with age. As we stated in the discussion (page 11, lines 27-31), in contrast to our results, decreased lamin B levels are found in brains from Alzheimer and Parkinson's disease patients bearing the LRRK2(G2019S) mutation (Frost et al., 2016, *Curr Biol* 26:129; Liu et al., 2012, *Nature* 491:603), and in aged primary human fibroblasts (Freund et al., 2012, *Mol Biol Cell* 23:2066) and keratinocytes (Dreesen et al., 2013, *J Cell Biol* 200:605), being a marker of cellular senescence. Thus, this data suggests that aging could course with decreased levels of lamin B1. However, to rule out that increased lamin B1 levels could be related with the aging process itself, we analyzed the correlation between lamin B1 levels and age in human samples from the putamen, cortex and hippocampus. Results showed that there is no correlation between age and lamin B1 levels. In addition, we compared the distribution of ages in controls, HD patients from VS I-II, and HD patients from VS III-IV and observed a similar distribution between groups. With this data, we are confident in assuring that the alterations we observe regarding lamin B1 levels in HD patients are occurring because of the pathology rather than a matter of age, since the control and HD samples are age matched. These results have been included in the revised version of the manuscript as Appendix Figure S1. In addition, and following the referee's suggestion, we have analyzed full z plane in the putamen of non-affected individuals and HD patients (an average of 50 nuclei/sample) in order to discard that nuclear invaginations are only occurring because of the aging process. Our results show that nuclear morphology of MSNs is altered in the putamen of VS III-IV HD patients. These results have been included as Figure 4D in the revised manuscript. Results (page 3, lines 29-33; page 4, lines 1-3) and discussion (page 11, lines 31-32) sections have been modified accordingly (text highlighted in yellow).

3. The mouse histology images presented are not convincing. It is very clear that the images presented are often from different depths within the mouse brain based on the morphology of the DG. This makes interpretation extremely complicated. The zoomed panels of neuronal nuclei make it increasingly clear that the authors are imaging vastly different z depths in their HD animals compared to controls. In many images, the control seems to be taken from a central section of a nucleus (where one can see a nice lamin ring). However, in the HD mice, the images appear to be from the top or bottom face of nuclei as one can observe the normal folds of the neuronal nuclear lamina. This is supported by vastly different NeuN intensities. Moreover, zoomed images of the CA1 region often show a different lamin intensity than that in the overview image- can the authors please reconcile this?

As indicated in the Methods and Results sections, mouse striatal and hippocampal images displayed in Figure 2a and 2c (initial submission, now Fig 2A and 3A) were obtained from 30 μm brain sections by confocal microscopy by covering the whole z-axis, inter-spaced by 0.5 μm from one to the other. Being that said, we cannot

understand how reviewer 3 conceives that: ***“In many images, control seems to be taken from a central section of a nucleus (where one can see a nice lamin ring). However, in the HD mice, the images appear to be from the top or bottom face of nuclei as one can observe the normal folds of the neuronal nuclear lamina.”***, considering that nuclei within striatal or hippocampal tissue will, of course, not perfectly align in a single Z plane, but rather every Z plane will contain nuclei aligned in a different relative Z coordinate. To clarify this point, we have included an example of sequential images of the different Z-planes comprised in the recorded Z-stacks as well as 3D surface reconstructions of wild-type and R6/1 mice CA1 hippocampal region, evidencing the clear increase in nuclear folds and nuclear elongation found in R6/1 mice neurons (Figure 3B in the revised manuscript). Results section has been rewritten accordingly (page 5, lines 7-8; text highlighted in yellow).

In addition, the commented ***“vastly different NeuN intensities”***, which we understand makes reference to merge images of figure 2a (initial submission, now Fig. 3A), is rather a visual effect of the overlapping nucleoplasmic lamin B1 red signal than a decrease in the NeuN (green) intensity since this effect is not appreciable in the separated NeuN green channel image on the same panel.

4. From both the histology and the fluorescence activated nuclear suspension imaging, it is unclear whether these "folds" and invaginations are truly pathological. The authors appear to present a single Z slice which is very misleading-and inappropriate when examining the 3d structure of the nucleus. Moreover, there are normally many folds in the neuronal nuclear lamina. Therefore, it is unclear if the authors were to examine more nuclei if these phenotypes would truly represent pathology. The authors must examine the full volume of the nucleus and not simply cutting thru the middle - as that will miss surface folding

When this study was started, and still nowadays, morphological analysis of highly compacted nuclear preparations, as the ones found in brain tissue (i.e. DG and CA-1 hippocampal regions), represented a huge challenge due to difficulties on performing proper individual nuclear segmentation. For this reason, we decided to develop the FANSI approach as an alternative for nuclear morphology quantification. Reasoning that, if nuclear morphology is maintained through the purification process thanks to the initial cross-linking step, the three-dimensional random orientation of nuclei recorded in individual events by the ImageStream flow cytometer microscopy system should capture population effects on morphological parameters, as we actually demonstrate. Although image acquisition is not based on confocal imaging, “Shape change” wizard in IDEAS® Software allows us to measure the circularity of any population by creating an analysis template. Figure below (Referee #3 Figure 2) describes how circularity is measured by IDEAS® Software and an example of low and high circularity events identified by “Shape change” wizard. Finally, we would like to highlight that for mice experiments, every biological replicate is the result of the acquisition of a minimum of 10,000 single events.

Circularity Feature

This feature measures the degree of the mask's deviation from a circle. Its measurement is based on the average distance of the object boundary from its center divided by the variation of this distance. Thus, the closer the object to a circle, the smaller the variation and therefore the feature value will be high. Vice versa, the more the shape deviates from a circle, the higher the variation and therefore the Circularity value will be low. See also .

Referee #3 Figure 2. Description of circularity measurement and example of low and high circularity events identified by “Shape change” wizard. Obtained from IDEAS® Software User’s Manual. Version 6.0, March 2013 ([https://pedsresearch.org/uploads/pages/img/IDEAS_User_Manual_6.pdf; pages 43 and 198](https://pedsresearch.org/uploads/pages/img/IDEAS_User_Manual_6.pdf;pages_43_and_198))

Being that said, and as commented in point 3, histological images displayed in Figure 2a and 2c (left panel, initial submission; now Fig. 2A and 3A) are maximal Z-projections of confocal images covering the whole Z-plane of a 30 μm brain section, and not a single Z-plane. We have tried to conduct additional analysis of morphological parameters using confocal images from mouse histological sections and recently developed ImageJ tools for improved nuclear segmentation (Stardist, <https://arxiv.org/abs/1806.03535>) and morphological assessment analysis (MorpholibJ, PMID 27412086) with success in the case of the striatum (results included in the revised version of the manuscript as Figure EV4). However, nuclear segmentation in the hippocampus was not precise due to the higher level of nuclear overlap and altered morphology found in R6/1 mice (see Figure 3B in the revised manuscript) resulting in poor nuclear segmentation performance. Methods (page 19, lines 18-21 and 25-31) and results (page 4, lines 31-33) sections have been modified accordingly (text highlighted in yellow).

5. The dextran assay as described is not an appropriate assessment of passive NCT and NPC permeability. The protocol described uses Triton for nuclear isolation. Triton alone can effectively permeabilize nuclear pores, thus allowing dextran to leak in. Nuclei should be isolated without Triton. Further, the authors must evaluate additional size dextrans (10kD, 70kD, 500kD) to support any conclusions regarding the permeability of the NPC. Moreover, given the appropriate experimental set up, FRAP is not needed to evaluate the passive permeability properties of the nucleus.

We perfectly understand the reviewer’s concern about the use of Triton for the isolation of nuclei as this was our concern too. Before performing the experiments shown in Figure 4 (initial submission, Figure 6 in the revised manuscript) we isolated nuclei with and without 0.1 % Triton X-100 (the dose used for the FRAP experiments). Referee #3 Figure 3 shows that the use of this dose of Triton for the isolation of nuclei does not affect the permeability of 20kDa- FITC-dextran. Thus, we consider that the conclusions from our permeability experiments are solid and we did not repeat the experiments without using triton.

Referee #3 Figure 3. The use of 0.1% Triton X-100 for nuclear isolation does not affect their permeability to 20kDa- FITC-dextran. Striatal nuclei from 30-week-old wild-type mice were isolated without (- T) or with (+ T) Triton and incubated with 20kDa- FITC-dextran. The nuclear permeability was analyzed by using FRAP. The half time of recovery (seconds) is shown. Each point corresponds to the value from an individual nucleus. Bars represent mean \pm S.E.M. Two-tailed unpaired Student's t test.

Regarding the use of FRAP as an appropriate method for studying passive permeability, we agree with the reviewer that many studies evaluate passive permeability by analyzing the entrance of different size dextrans into the nucleus without using FRAP. However, these studies evaluate a disruption in passive permeability originated by a leaking nuclear membrane. Thus, they can easily identify an increase in nuclear fluorescence or the incorporation of ≥ 70 kDa dextrans (Roehrig et al., 2003, Anal Biochem 318:244; Strasser et al., 2012, Nucleus 3:530; Eftekharzadeh et al., 2018, Neuron 99:925), both being a sign of leaking membranes and nuclear permeability dysfunction. However, our hypothesis was contrary to this. We expected a more rigid and clogged membrane due to the increase in lamin B1 that would lead to decreased incorporation of dextran molecules into the nucleus. Thus, in order to face this question, we could not use ≥ 70 kDa dextrans, since it has been widely reported that they do not incorporate into the nucleus unless the membrane is leaking (Roehrig et al., 2003, Anal Biochem 318:244; Strasser et al., 2012, Nucleus 3:530; Eftekharzadeh et al., 2018, Neuron 99:925). Hence, we had to use small size dextrans. We first analyzed nuclear permeability by using 40 kDa- FITC-dextran and found that it was barely incorporated in R6/1 striatal nuclei indicating a dysfunction in nuclear permeability for ≥ 40 kDa molecules (Referee #3 Figure 4).

Referee #3 Figure 4. Incorporation of 40kDa- FITC-dextran is reduced in striatal R6/1 mouse nuclei. Striatal nuclei from 30-week-old R6/1 mice were incubated during 10 min. with 40kDa- FITC-dextran and fluorescence incorporated to the nucleus was analyzed. Bars represent mean \pm S.E.M. Two-tailed unpaired Student's t test.

To further understand nuclear permeability dysfunction in HD nuclei, we decided to use 20 kDa dextran and explore nuclear permeability more accurately by using FRAP. This technique allowed us not only to analyze the total incorporation of dextran, but also the incorporation rate. This last parameter was of high interest for us since we hypothesized that increased lamin B1 levels could be acting as an impediment to the entry of molecules to the nucleus rather than causing a total blockage. This is what we actually observed since 20 kDa dextran incorporation rate was slower in R6/1 mouse nuclei with increased levels of lamin B1 (striatal MSNs and hippocampal CA1 neurons nuclei) than

in wild-type mouse nuclei. Therefore, FRAP allowed us to identify an alteration in dextran incorporation rate in HD nuclei, an important finding enlightening one of the functional consequences of lamin B1 increase. We think we have efficiently proven the link between nuclear permeability dysfunction and lamin B1 alterations since nuclear permeability was only improved in those neurons where lamin B1 levels were normalized (Figure 10G in the revised manuscript). Thus, we consider that FRAP is an appropriate set up for addressing our hypothesis.

6. Overall, the authors fail to provide any conclusive evidence that increased lamin levels contribute to the pathogenesis of HD. Their drug studies result in incredibly subtle changes in lamin levels. Moreover, the main readout from these studies is mouse behavior. However, behavioral rescues could be independent of the subtle effects on lamin levels. These studies are in no way tied back to the molecular defects presented throughout the course of this manuscript. The authors should evaluate the effects of lamin OE in primary neuronal cultures to determine if molecular alterations are reminiscent of those seen in HD. Moreover, does KD of lamin in primary HD neuronal cultures mitigate relevant phenotypes?

We agree with the reviewer that treatment with betulinic acid induces subtle changes in lamin B1 levels and therefore, beneficial effects could be related to the regulation of other mechanisms (see the response to concern 5, Referee #1). However, we observed that normalization of lamin B1 levels in R6/1 mouse CA1 hippocampal neurons nuclei after betulinic acid treatment was accompanied by a partial rescue of nuclear morphology alterations and an amelioration of nucleocytoplasmic transport dysfunction. Moreover, betulinic acid treatment did not normalize lamin B1 levels in R6/1 mouse striatal MSNs nuclei, which still presented morphological alterations and transport dysfunction (Supplementary Figure 7 in the initial submission, Appendix Figure S12A and B in the revised manuscript). Therefore, we think that altogether these results suggest that the alteration in lamin B1 levels is involved in nuclear dysfunction in R6/1 mouse.

To better respond to the reviewer's concern, and following his/her suggestion, we overexpressed lamin B1 in primary striatal neurons and observed that those neurons overexpressing lamin B1 showed alterations in nuclear morphology (Figure 5 in the revised manuscript) thus resembling what we observed in the nucleus of R6/1 mouse striatal MSNs and CA1 hippocampal neurons. Material and methods (page 17, lines 30-32; page 18, lines 1-5; page 19, lines 1-10, 18-22 and 30-32) results (page 6, lines 12-18; page 11, lines 31-32) and discussion (page 13, lines 8-10) sections have been modified accordingly (text highlighted in yellow).

Moreover, as also suggested by the Referee, we decided to evaluate the effect of lamin B1 knockdown in primary cultures from R6/1 mouse as we agree with the referee that this experiment would directly demonstrate the importance of lamin B1 increase for neuron nucleus dysfunction in HD. However, we had a set of experimental problems as described below.

1.- To knock-down lamin B1, we acquired a collection of short-hairpin RNAs from the RNAi consortium (TRC) consisting in 5 different sequences specifically designed to target mouse *LMNB1* mRNA and a scramble control codified in the pLKO.1-puro vector. This vector contains a puromycin resistance gene for mammalian stable selection of transfected cells that was already tested in our lab, resulting in a nice and stable reduction of the target gene (Creus-Muncunill, Rué et al., 2018, Mol Neurobiol 55:7728). However, to our surprise, after transfection in the STHdh^{Q7/Q7} striatal cells and puromycin selection during 72h, lamin B1 levels were not decreased in any of the five different short-hairpin RNA sequences used (Referee #3 Figure 5).

Figures for Referee not shown.

2.- In parallel, we analyzed lamin B1 levels by Western blot in wild-type and R6/1 mouse striatal primary cultures at 14 days *in vitro* (DIV). At this time point, we did not detect alterations in lamin B1 levels in R6/1 when compared with wild-type mouse striatal primary neurons (Referee #3 Figure 6). We decided to stop the experiments here and do not examine other time-points (DIV), as we did not succeed in silencing lamin B1 levels.

Figures for Referee not shown.

7. The nuclear lamina is tightly linked to nuclear pores. In fact many nucleoporins actively coordinate genome organization and transcription. However, the authors assume all transcriptional alterations are a result of subtle and variable increases in lamin expression. This reflects a lack of scientific understanding of nuclear function and the many proteins and structures in the nuclear membrane- eg. The nuclear pore (and its 30 different proteins) or the LINC complex. It is quite cavalier, and very likely wrong, to assume that observed HD phenotypes (in particular those relating to membrane permeability and transcription) are solely the effect of increases in lamin. To be comprehensive, the authors would have to examine the changes in the nuclear pore structure (e.g the many nucleoporins) and the linc complex as well- as these may be altered in HD (as others have already published)

We agree with the reviewer that assuming that the permeability and transcriptional changes occurring in HD are due solely to alterations in lamin B1 levels is wrong. That was not our intention and we apologize if this was the impression. The purpose of this study has always been to state that changes in lamin B1 levels may participate in the pathology and among other mechanisms alterations in nuclear permeability and transcription seem to play a role. We also agree that examining the changes in the nuclear pore structure and LINC complex in HD would be very interesting, but this is out of the scope of the present work. As the reviewer points out, and we mention in the manuscript, previous work has already shown alterations in nuclear pore in HD and we hope our work could serve as evidence for more specialized people to analyze whether lamin B1 interactions with components of the nuclear pore structure or LINC complex are altered in HD. Indeed, nuclear pore protein TPR associates with lamin B1 and in cooperation they contribute to maintain the nuclear pore complexes distribution (Fiserova et al., 2019, Cell Mol Life Sci 76: 2199). Thus, further examination of the implications of increased lamin B1 on nuclear pore organization in HD would be of high interest.

8th Nov 2020

Dear Dr. Perez-Navarro,

Thank you for the submission of your revised manuscript to EMBO Molecular Medicine. We have now received the enclosed report from two referees who were asked to re-assess it. As you will see the referees are now supportive and I am pleased to inform you that we will be able to accept your manuscript pending the following amendments:

***** Reviewer's comments *****

Referee #1 (Remarks for Author):

The authors have responded to the comments of the reviewers by clarifying the relationship of the lamin changes they are seeing to age, revising the presentation of the human data, expanding analysis of different cell types, replacing images where needed, and performing preliminary studies to address the possible mechanism of lamin upregulation. These changes improve the manuscript.

Referee #2 (Remarks for Author):

The revised article includes new experiments and analyses and it is very much improved. Overall, the authors addressed all my concerns.

2nd Authors' Response to Reviewers

20th Nov 2020

The authors have made the requested editorial changes.

Accepted

24th Nov 2020

We are pleased to inform you that your manuscript is accepted for publication and is now being sent to our publisher to be included in the next available issue of EMBO Molecular Medicine.

Corresponding Author Name: Esther Pérez-Navarro

Manuscript Number: EMM-2020-12105